# LEARNED COARSE MODELS FOR EFFICIENT TURBULENCE SIMULATION

**Kimberly Stachenfeld**,[1]   **Drummond B. Fielding**,[2] **Dmitrii Kochkov**,[3]
**Miles Cranmer**,[4]   **Tobias Pfaff**,[1]   **Jonathan Godwin**,[1]   **Can Cui**,[2]
**Shirley Ho**,[2]   **Peter Battaglia**,[1] **Alvaro Sanchez-Gonzalez**[1]

[1]DeepMind, London, UK   [2]Center for Computational Astrophysics, Flatiron Institute, New York, NY
[3]Google Research, Cambridge, MA   [4]Princeton University, Princeton, NJ
`alvarosg@google.com, stachenfeld@google.com`

## ABSTRACT

Turbulence simulation with classical numerical solvers requires high-resolution grids to accurately resolve dynamics. Here we train learned simulators at low spatial and temporal resolutions to capture turbulent dynamics generated at high resolution. We show that our proposed model can simulate turbulent dynamics more accurately than classical numerical solvers at the comparably low resolutions across various scientifically relevant metrics. Our model is trained end-to-end from data and is capable of learning a range of challenging chaotic and turbulent dynamics at low resolution, including trajectories generated by the state-of-the-art `Athena++` engine. We show that our simpler, general-purpose architecture outperforms various more specialized, turbulence-specific architectures from the learned turbulence simulation literature. In general, we see that learned simulators yield unstable trajectories; however, we show that tuning training noise and temporal downsampling solves this problem. We also find that while generalization beyond the training distribution is a challenge for learned models, training noise, added loss constraints, and dataset augmentation can help. Broadly, we conclude that our learned simulator outperforms traditional solvers run on coarser grids, and emphasize that simple design choices can offer stability and robust generalization.

## 1 INTRODUCTION

Turbulent fluid dynamics are ubiquitous throughout the natural world, and many scientific and engineering disciplines depend on high quality turbulence simulations. Examples range from design problems in aeronautics (Rhie, 1983) and medicine (Sallam & Hwang, 1984) to scientific problems of understanding molecular dynamics (Smith, 2015), weather (Beljaars, 2003), and vast galactic systems (Canuto & Christensen-Dalsgaard, 1998). Predicting turbulent dynamics is challenging, and finding a general solution to the governing equations of fluid dynamics, the Navier-Stokes Equation, is a famously open problem in Mathematical Physics (Jaffe, 2006). The challenge is due in large part to the fact that turbulence is chaotic: small-scale changes in the initial conditions can lead to a large difference in the outcome. Nonetheless, over the past several decades, high-quality numerical solvers have been engineered that integrate the governing partial differential equations (PDEs) and can maintain accuracy over long integration periods even for complex dynamics. These solvers often must operate over high-resolution grids and therefore require substantial computational resources. Otherwise, high-frequency information is lost due to hard-to-model "numerical viscosity", and the simulated dynamics can further diverge from the underlying equations due to the chaotic dynamics.

Learned simulation represents a promising avenue for efficient fluid simulation because learned models can potentially adapt to capture the large-scale dynamics on coarse grids. Learned simulators vary in the extent to which they incorporate components from classical solvers, like those that learn closures and subgrid discretization (Duraisamy et al., 2019; Freund et al., 2019; Pathak et al., 2020; Um et al., 2021; Kochkov et al., 2021), versus adopting a pure Machine Learning (ML) approach (Kim et al., 2018; Lusch et al., 2018; Sanchez-Gonzalez et al., 2020; Wang et al., 2020; Li et al., 2020c;a; Pfaff et al., 2021). Advantages of the pure ML route are that the same model can be used

without specialized knowledge of the domain and that they do not involve the challenge of interfacing with PDE solvers. However, fully learned simulators often work well only when conditions are similar to the training distribution, which can limit their stability over time and their generalization capabilities.

Our primary contribution is to demonstrate that fully learned simulators can learn to predict turbulent dynamics on coarse grids more accurately than classical solvers. This is true over a range of performance metrics, but the learned simulators particularly excel in their ability to preserve high frequency information. While learned simulators can be prohibitively unstable, we show that the simple practice of tuning training noise and the temporal downsampling factor can reliably produce stable rollouts. We introduce a simple model that combines neural networks architectures for grids (Dilated CNNs (Yu & Koltun, 2016), ResNets (He et al., 2015)) with recent advancements on general purpose learned simulation on graphs (encode-process-decode models (Sanchez-Gonzalez et al., 2020; Pfaff et al., 2021) trained with noise (Sanchez-Gonzalez et al., 2018; 2020; Pfaff et al., 2021)). We implement this as well as a set of learned models from the literature that have been proposed for modeling turbulence, and show that Dilated ResNet (Dil-ResNet) outperforms the often more complicated, more heavily parameterized models. Rather than looking at a single problem, we evaluate our models over a variety of challenging turbulent and chaotic domains, showing that Dil-ResNet is generally able to capture a variety of turbulent dynamics out-of-the-box. This includes two environments generated by the new `Athena++` solver(Stone et al., 2020), a state of the art (magneto-)hydrodynamics simulator used for astrophysics, where extremely high-resolution and large-scale simulations are often required. Finally, we evaluate the ability of the learned models to generalize to initial conditions, rollout durations, and environment sizes outside of the training distribution. Generalizing out of distribution is a known challenge for learned models, and we document where our models fail and what model and training choices can improve generalization.

## 2 RELATED WORK

Turbulence simulation is a crucial step in problems across disciplines. These include forecasting problems, like predicting the spread of a wildfire (Leonardi et al., 2019) or the evolution of a hurricane (Zhang et al., 2020), engineering challenges, like designing more aerodynamic machinery (Rhie, 1983), and scientific questions, like understanding the physics of insect flight (Dickinson et al., 1999) or dynamics of celestial objects (Fielding et al., 2018). Across applications, a variety of statistical measures are relevant for quantifying physical variables under turbulence (Velocity Autocorrelation, variable histograms, Cooling, Energy spectra). The primary challenge for simulating turbulence is accurately capturing small- and large-scale flow dynamics, and their interplay, simultaneously.

Traditional approaches to computational fluid dynamics integrate the governing PDEs describing the fluid flow, given initial and boundary conditions. State-of-the-art turbulence simulators, such as `Athena++`, are built upon efficient algorithms that prioritize the preservation of high-order statistics. Such simulators are used frequently in scientific applications to numerically compare model predictions with experimental observations, but can be extremely–often prohibitively–expensive to run. Thus, there is significant practical interest in developing more efficient simulators.

Deep learning models for predicting physical dynamics have traditionally been used for modeling low-dimensional dynamical systems with graph-like (Battaglia et al., 2016; Sanchez-Gonzalez et al., 2018) and grid-like specifications (Raissi et al., 2017; Chen et al., 2018), but there is a growing body of work for fine-grained modeling of complex dynamics, as in particle-based liquids and soft-body physics (Li et al., 2019; Ummenhofer et al., 2020; Sanchez-Gonzalez et al., 2020). Learned models are often used in computer graphics to speed up (Wiewel et al., 2019; Um et al., 2018; Ladický et al., 2015; Holden et al., 2019) or super-resolve (Xie et al., 2018) simulations. A number of papers predict the steady-state flow around 2D airfoil wings, using CNN (Thuerey et al., 2020; Bhatnagar et al., 2019) or GCN (Belbute-Peres et al., 2020) architectures, or the transient dynamics of various low Reynolds number flows using mesh representations (Pfaff et al., 2021).

In recent years, a number of models have been introduced specifically for learning turbulent dynamics. Many of these use neural networks to model some part of an otherwise classical, theoretically motivated numerical integration setup (Duraisamy et al., 2019; Freund et al., 2019; Pathak et al., 2020; Um et al., 2021; Kochkov et al., 2021; Tompson et al., 2016), PDE formulation (Raissi et al., 2017; Champion et al., 2019), or embedding space (Lusch et al., 2018). Other models incorporate

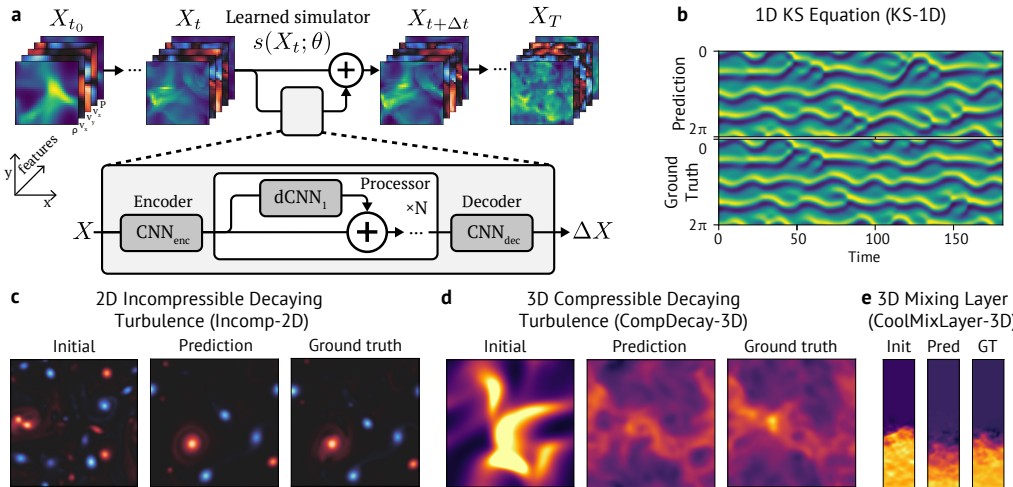

Figure 1: (a) Learned simulator framework. The model is trained to predict the difference between the current and next state. Gaussian noise can be added to the inputs during training to produce models that are robust to small perturbations. Shown below is the schematic for the Dilated ResNet model (Dil-ResNet). (b-e) Frames predicted by rollouts from the learned simulator model compared to ground truth frames. (b) KS-1D: The model follows the ground truth closely for the first 150 steps (t < 75) and remains plausible thereafter. (c) INCOMP-2D: The model remains accurate after 119 model steps (91,392 solver steps). (d) COMPDECAY-3D: The model remains accurate after 31 model steps (1984 solver steps). (e) COOLMIXLAYER-3D. The model remains qualitatively accurate after 59 model steps (59,000 solver steps) of a box size of ($L = 0.75$). Videos available at sites.google.com/view/learned-turbulence-simulators.

some information about turbulence simulation into the architecture, but like ours, are learned end-to-end in a supervised way. Wang et al. (Wang et al., 2020) implemented a fully learned, neural network "Turbulent Flow Network" (TF-Net) that applies physically inspired decomposition with learned parameters and processes the components with U-Nets for fluid simulation. Li and colleagues introduced "Fourier Neural Operators" for turbulence simulation (Li et al., 2020a), which apply neural networks that convolution parameters in the Fourier domain. This model is part of a broader family of work (Portwood et al., 2019; Li et al., 2020b) which combines basis transforms (Fourier, Multipole, etc) with neural operators that process the transformed input. We implement models from these papers as baselines to compare to our learned simulator.

## 3 MODEL

### 3.1 LEARNED SIMULATION FRAMEWORK ON GRIDS

We train a simulation model $s : \mathcal{X} \rightarrow \mathcal{X}$ that maps $X_t$ to $X_{t+\Delta t}$, where $\mathcal{X}$ is a space of $n$+1 dimensional tensors which represent the state variables for each point on a grid with $n$ spatial dimensions and one feature dimension at time $t$. The true physical dynamics applied over $K$ time steps yields a trajectory of states which we denote $(X_{t_0}, ..., X_{t_K})$. The "rollout" trajectory produced by iteratively applying $s$ for $K$ steps is denoted $(\tilde{X}_{t_0}, \tilde{X}_{t_1}, ..., \tilde{X}_{t_K})$, where $\tilde{X}_{t_0} = X_{t_0}$ represents initial conditions given as input. Our learnable simulators (Fig. 1a) implement $s$ using neural networks denoted NN (with weights $\theta$), and predict the next state as the difference from the current state, $\tilde{X}_{t+\Delta t} = s(\tilde{X}_t; \theta) = \tilde{X}_t + \text{NN}(\tilde{X}_t; \theta)$.

### 3.2 MODELS

In this work we study "fully learned" models, which learn the physical dynamics end-to-end without any hard-coded solver components. We implemented a novel Dilated ResNet model, as well as several baseline models. Detailed model parameters can be found in the Appendix.

**Dilated ResNet Encode-Process-Decode (Dil-ResNet)** We designed this architecture to combine the encode-process-decode paradigm (Sanchez-Gonzalez et al., 2018; 2020) with dilated convolutional networks (Fig. 1a). The encoder and decoder each consist of single linear convolutional layer. The processor consists of $N = 4$ dilated CNN blocks (dCNN$_n$) connected in series with residual connections (Yu & Koltun, 2016; He et al., 2015). Each block consists of 7 dilated CNN layers with dilation rates of (1, 2, 4, 8, 4, 2, 1). A dilation rate of N, indicates each pixel is convolved with pixels that are multiples of N pixels away (N=1 reduces to regular convolution). This model is not specialized for turbulence modeling *per se*, and its components are general-purpose tools for improving performance in CNNs. Residual connections help avoid vanishing gradients, and dilations allow long-range communication while preserving local structure. All individual CNNs use a kernel size of 3 and have 48 output channels (except the decoder, which has an output channel for each feature). Each individual CNN layer in the processor is immediately followed by a rectified linear unit (ReLU) activation function. The Encoder CNN and the Decoder CNNs do not use activations.

**Encode-U-Net-Decode (U-Net)** We designed this architecture as a simplification of the TF-Net baseline explained below. It uses a simple linear CNN encoder and decoder (same as Dil-ResNet) and a standard U-Net block (Ronneberger et al., 2015). As in (Ronneberger et al., 2015), each CNN stack is made of 3 CNNs, followed by activations. Other hyperparameters of the U-Net also matched those used in (Ronneberger et al., 2015).

**Turbulent Flow Net (TF-Net) (Wang et al., 2020)** TF-Net uses a domain-specific variation of U-Nets to model turbulence, along with other architectural elements inspired by RANS-LES Hybrid Coupling in the encoder. Hyperparameters matched those used in (Ronneberger et al., 2015; Wang et al., 2020).

**Constrained Turbulent Flow Net (Con-TF-Net) (Wang et al., 2020)** Wang et al. also explored a constraint term in the loss which penalizes the divergence, which was constrained to be zero everywhere for the PDE in their incompressible turbulence experiments. We implemented a similar model, adapting the constrained function to match constraints or preserved quantities in our datasets. For KS-1D, the mean is constrained to be 0; for INCOMP-2D, the divergence is constrained to be 0, and for COMPDECAY-3D the total energy is constrained to be the same as input's total energy.

**Constrained Dilated ResNet (Con-Dil-ResNet)** This model is Dil-ResNet (described above) with the additional constraint terms from (Con-TF-Net) added to the loss function. This allows us to examine the effect of the loss term separately from the other aspects of the TF-Net architecture.

**Fourier Neural Operator (FNO) (Li et al., 2020a)** This model uses the neural operator formalism introduced in earlier work (Li et al., 2020c;b), applied in the Fourier domain. We implemented FNO for 1D and 2D spatial domains as in (Li et al., 2020a) Hyperparameters are from (Li et al., 2020a).

**CNN padding** We use periodic padding for spatial axes to implement periodic boundary conditions. For the COOLMIXLAYER-3D, which has a fixed boundary condition along the vertical axis, we mask out the value at the boundary when computing training loss and set it to the ground truth at each step of the rollout. We also augment the input state with a binary feature to distinguish boundary pixels.

## 3.3 TRAINING

**Loss** All models[1] are trained to predict $\Delta X$. We use a mean square error loss $\ell(X_t, X_{t+\Delta t}) = \text{MSE}(\text{NN}(X_t; \theta), \Delta X)$ to optimize parameters $\theta$. Input $X_t$ and target $\Delta X = X_{t+\Delta t} - X_t$ features are normalized to be zero-mean and unit variance, which seemed to improve training speed. The additional mean square error loss components for constraint preservation in Con-Dil-ResNet and Con-Dil-ResNet are added to the main loss with a relative weight of 1.

**Training noise** Optionally, we trained with Gaussian random noise with fixed standard deviation $\sigma$ added to the input $X_t$ and subtracted from the target $\Delta X$, as this has been shown to improve stability of rollouts and prevent error accumulation by training the model to correct for small errors (Sanchez-Gonzalez et al., 2018; 2020; Pfaff et al., 2021). See Appendix for more details.

---

[1]In the original papers, TF-Net, Con-TF-Net, and FNO directly predict $X$ but we did not find substantive differences from predicting $\Delta X$. By having all models predict $\Delta X$ as a target, this allows a fairer comparison by reusing the same target normalization.

## 4 EXPERIMENTAL TURBULENT DOMAINS

We use four PDEs exhibiting chaotic or turbulent dynamics representative of a wide range of problems across science and engineering. (Fig. 1b-e). These environments range from the classic problems (KS Equations, Incompressible Turbulence) to more challenging environments that incorporate compressibility and non-uniform initial conditions. The training data is generated at high resolution Spatial and temporal downsampling factors and additional details about the environments are shown in the Appendix.

**1D Kuramoto-Sivashinsky Equation (KS-1D):** A PDE that generates unstable, chaotic dynamics in 1D, solved using Fourier spectral method (Kuramoto, 1978; Sivashinsky, 1977). The state consists of a scalar velocity $v$ per 1D grid point. While KS-1D is not technically turbulent, it is a well-studied chaotic equation that is useful for assessing the ability of our learned models to capture dynamics that are highly unstable and nonlinear.

**2D Incompressible Decaying Turbulence (INCOMP-2D):** Fluid flow modeled by Navier-Stokes in which small-scale eddies decay into large-scale structures due to the inverse energy cascade, solved using direct numerical simulation (Kochkov et al., 2021). The state consists of a vector velocity $v_x, v_y$ per 2D grid point. These simulations are relevant to open questions in atmospheric dynamics (Boffetta & Ecke, 2012a).

**3D Compressible Decaying Turbulence (COMPDECAY-3D):** Decaying transonic turbulent flow under Navier-Stokes, assuming adiabatic equation of state with adiabatic index $\gamma = 5/3$. The state consists of scalar density $\rho$, vector velocity $v_x, v_y, v_z$, and scalar pressure $P$ per 3D grid point. Energy at each grid point is $E = 1/2\rho v^2 + 3/2P$. Simulations were carried out with `Athena++` (Stone et al., 2020), a state-of-the-art simulator used in astrophysics. All real fluids are somewhat compressible, and this is particularly non-negligible when modeling gases (Pope, 2000). In astrophysics problems, for which the `Athena++` was developed, understanding the properties of these flows plays a crucial role in regulating planet, star, black hole, and galaxy formation (Brandenburg & Nordlund, 2011).

**3D Compressible Turbulence with Radiative Cooling Mixing Layer (COOLMIXLAYER-3D):** Turbulent mixing resulting from the Kelvin-Helmholtz instability, caused by velocity differences across the interface between fluids of different densities, solved with `Athena++`. The state is specified in the same way as in COMPDECAY-3D. Mixing involves strong cooling, leading to net flow from the low-density phase into the mixing layer. This environment was additionally challenging because of its open boundaries and non-uniform initial conditions. This process is common in atmosheric flows as well as many aspects of galaxy formation (Fielding et al., 2020).

## 5 RESULTS

We evaluate the learned simulator models in terms of stability, performance, efficiency, and generalization. Unlike numerical solvers, which are PDE specific, learned simulator models can be designed to be reusable across domains. In Figure 1, we show that the same general-purpose architecture (Dil-ResNet) and loss, trained on each of the domains, learns to capture a range of qualitatively diverse turbulent dynamics across domains (Fig. 1b-e, videos: sites.google.com/view/learned-turbulence-simulators). All results are evaluated on held-out test trajectories sampled from the same distribution used for training (except for generalization sections). All units (except time) are normalized and therefore dimensionless.

**Spatial coarsening** Numerical solvers are known to lose information, particularly high-frequency information, when applied to coarse grids. We compared our Dil-ResNet learned simulator to `Athena++` run on spatially coarsened grids (see Section 5 for comparison with learned models). We chose the energy field $E = \frac{1}{2}\rho v^2 + \frac{3}{2}P$ as the quantitative metric of 3D turbulence for the main text, as it summarizes performance on all state variables. We show results for `Athena++` run at resolutions of $32^3$ (the same resolution as the learned models) and at $64^3$ (higher resolution than the learned models, lower than the ground truth). The "ground truth" data is from `Athena++` run at $128^3$. All grids are downsampled to $32^3$ for comparison. We compared models in terms of the RMSE

---

[3]Note the reason that error rises and falls for this model is that COMPDECAY-3D is decaying turbulence. As time passes, the distribution becomes more uniform in space, resulting in lower error across models.

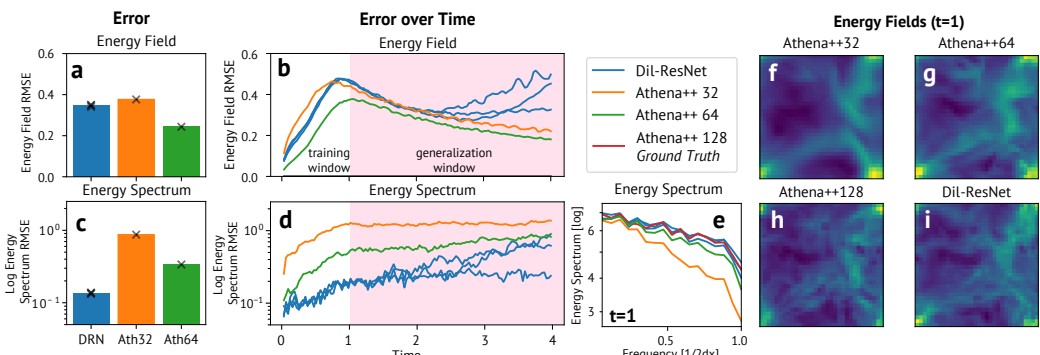

Figure 2: Comparison between learned $32^3$ Dil-ResNet and same-resolution $32^3$ `Athena++`, intermediate resolution $64^3$ `Athena++`, and ground truth high resolution $128^3$ `Athena++` in COMPDECAY-3D. (a) Energy Field RMSE ($y$-axis) for Dil-ResNet ($32^3$ resolution (blue)) and `Athena++` ($64^3$ (dark gray), and $32^3$ (light gray)) on a test trajectory the window of times seen during training. Dil-ResNet RMSE is lower than that of the comparably coarse `Athena++` $32^3$ but higher than `Athena++` $64^3$. (b) Energy Field RMSE ($y$-axis) as a function of rollout duration ($x$-axis) over the training window (white background) and generalizing beyond (pink background)[3]. The learned simulator's error grows faster starting around 3x the training window (at $t \approx 3000$). Each blue line corresponds to a different seed. Ground truth (red) error is at 0 and hidden by $x$-axis. (c) Energy Spectrum RMSE ($y$-axis). Dil-ResNet has lower error than both comparable and higher resolution $32^3$ and $64^3$ `Athena++` models. (d) Energy Spectrum RMSE ($y$-axis) over rollout. Dil-ResNet's rollouts maintain lower Energy Spectrum error for up to $4\times$ the training duration. (e) Energy Spectrum for the learned and `Athena++` simulators at rollout $t \approx 1$ s (the end of the training window). The $x$-axis represents spatial frequency, and the $y$-axis represents spectral power. Compared to Dil-ResNet, the coarse `Athena++` models lose power in the high frequency range. (f–i) Sample energy Fields at $t \approx 1$ s for Dil-ResNet and the three `Athena++` resolutions. States from coarse `Athena++` resolutions lose high frequency detail with respect to the ground truth, which the coarse learned model captures.

of the predicted Energy Field and the Log Energy Spectrum (See Appendix for definition). We used the energy field as the main variable as it is a combination of all five state variables, but see Videos and Appendix for per-variable results.

The learned simulator outperforms the comparable resolution `Athena++` $32^3$ across a variety of metrics, despite having no built-in specializations for turbulent dynamics. In Figure 2a-b, we show RMSE for the Energy Field for $t < 1$, which corresponds to the initial phase of the turbulence decay seen during training (white window in Fig 2b). Energy Field is given as $E = \frac{1}{2}\rho v^2 + \frac{3}{2}P$, and it implicitly summarizes performance on all state variables (other metrics are shown in the Appendix). The learned simulator outperforms both the same- and higher-resolution `Athena++` rollouts in terms of the Log Energy Spectrum (Fig. 2c,d), as the `Athena++` simulators lose high frequency components that the learned simulators preserve (Figure 2e-i). Log Energy Spectrum is computed by (1) taking the 3-D Fourier transform, (2) computing the amplitude of each component, (3) taking the histogram of the 3-D Fourier transform over frequency amplitude ($\sqrt{k_x^2 + k_y^2 + k_z^2}$) to get a 1-D PSD, and (4) then taking the log. We looked at a range of other physically relevant metrics and find that the learned simulator outperforms the comparably coarse $32^3$ `Athena++` but not $64^3$ `Athena++` on predicting feature histograms, the phase histograms (pressure v. density and entropy v. pressure), and mean squared error for each feature. The learned simulator outperformed both $32^3$ and $64^3$ `Athena++` simulators on higher order velocity autocorrelations, as well as spectrum error. Since different scientific questions rely on different metrics, the tradeoffs of learned versus physics based simulators may vary across applications. These metrics are defined in the Appendix.

**Stability and training noise** While scientific simulators are typically designed to be stable over time, a common failure mode in learned models is that small errors can accumulate over rollouts and lead to a domain shift. One reason for this is that, as the model is fed its most recent prediction back in as input for predicting future steps, its distribution of input states begins to deviate from

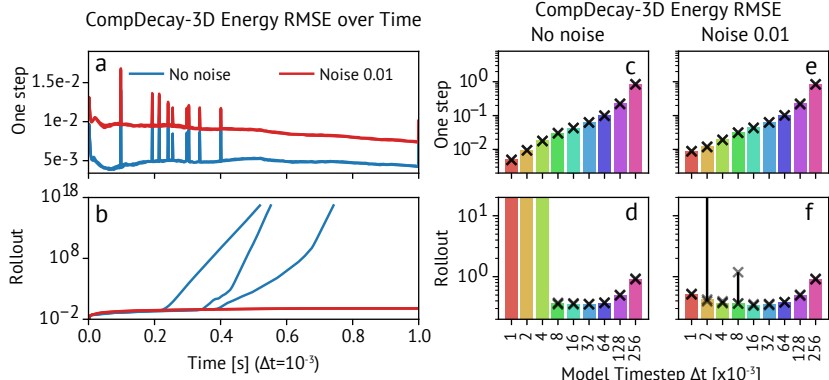

Figure 3: Effects of noise and temporal downsampling on rollout stability. (a) One step errors are larger for models trained with noise. Note the error spikes are very small and are not model-related general artifacts, but specific to particular frames of this test trajectory. (b) However, models trained without noise can yield unstable rollouts, especially when using very small time steps, which is not a problem for models trained with noise. (c, e) One-step model error rises monotonically with coarser temporal downsampling. (d, f) Rollout error has a U-shaped curve over temporal downsampling factors, for a trajectory of the same time duration, with minimum error around $\Delta t = 0.032$.

that experienced at training, where it fails to generalize and can make arbitrarily poor predictions. We found that adding Gaussian noise $\sigma = 0.01$ to the inputs $X_t$ during training led to less accurate one-step predictions (Fig. 3a), but more stable trajectories (Fig. 3b). This is of particular importance for models that take a very large number of small steps. This is presumably because the training distribution has broader support and the model is optimized to map deviant inputs back to the training distribution.

**Temporal coarsening** An advantage of learned simulators is that they can exploit a much larger step size than the numerical solver, as they can discover efficient updates that capture relevant dynamics on larger timescale. This allows for faster simulation. See Videos and Appendix for qualitative examples of Dil-ResNet trained on a large range of exponentially increasing coarse timesteps in KS-1D, INCOMP-2D and COMPDECAY-3D, which the model can adapt to (note that a separate model is trained for each $\Delta t$). Quantitatively, though the one-step error is at its lowest when using smaller time steps (Fig. 3c), the rollout error has an optimal time step at around 0.032 (Fig. 3d). This demonstrates the tradeoff between large and small time steps. Large time steps ($> 0.032$) cause predicting the next state to become more challenging. Small time steps ($< 0.008$), which require more simulator steps for the same duration, often yield unstable models because they provide more opportunities for errors to accumulate (Fig. 3d) (e.g. for some $\Delta t$, $1s/\Delta t$ steps are required). However, they can still be stabilized to some extent with training noise (Fig. 3f).

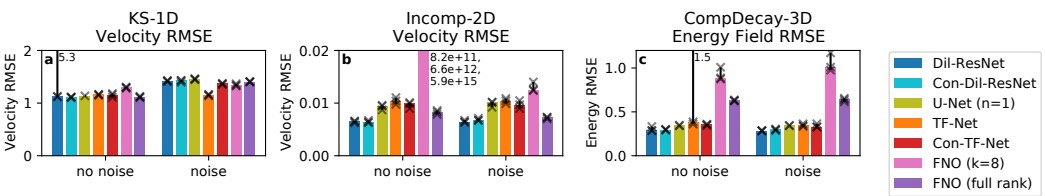

Figure 4: Comparison across learned models, contrasting noise and no-noise training conditions, across the three primary tasks (a) KS-1D, (b) INCOMP-2D, and (c) COMPDECAY-3D. With a few exceptions, the various learned models had comparable performance, though the Dilated ResNets (Dil-ResNet, Con-Dil-ResNet) consistently have the lowest error. In KS-1D (a), the noise harmed performance; in INCOMP-2D (b) the noise particularly benefits the FNO rollouts; in COMPDECAY-3D (c) the noise mainly stabilized rollouts.

**Comparison across learned models** In Figure 4 we compare the learned models in terms of the performance across the different primary turbulence datasets. We find that Dilated ResNets (Dil-ResNet, Con-Dil-ResNet) are the top performing models, numerically outperforming more specialized architectures. However, we find that all learned models (ours and the baselines) produce qualitatively good rollouts compared to `Athena++` $32^3$, suggesting learned simulators are generally a good approach to turbulence simulation on coarse grids (see Videos). We also find training noise improves stability across learned models. All models shown in Figure 4 are evaluated such that the first predicted step is the 7th step of the ground truth trajectory. This is to allow comparison with TF-Net, which takes $C = 6$ context states as input. We show in further detail the predictions of the different models in the Appendix.

**Running time** Learned simulators can accelerate simulations not only by coarsening in time and space, but also by running off-the-shelf on specialized GPU hardware, unlike e.g., `Athena++` which is currently CPU specific. By downsampling in time ($\Delta t = 0.0005 \rightarrow 0.032$) and space ($128^3 \rightarrow 32^3$), the Dil-ResNet model trained on COMPDECAY-3D running on a single GPU speeds up the wall-clock simulation time by 1000x compared to `Athena++` running on an 8-core CPU (further details in Appendix). GPU solvers can be developed for many applications and can greatly accelerate simulation; however, these can require a lot of expertise and time to develop, and will still require high spatial and temporal resolution grids.

**Constraint satisfaction and preserved quantities** Traditional solvers often implement constraints to preserve conserved quantities. Learned simulators do not necessarily learn such constraints and may fail to capture these (see Appendix). We found that training with noise prevents this in some cases: it helps Dil-ResNet keep the mean value of the KS-1D velocity, and the INCOMP-2D divergence bounded near zero. However, in COMPDECAY-3D using noise does not seem to help preventing the model from predicting drifts in total mass, vector momentum and energy, even though they are fixed across the whole dataset. We speculate this is because these preserved quantities only account for 5 degrees of freedom out of a 3D state that is made of $163840 (= 5 \cdot 32^3)$ output variables.

**Generalization to longer rollouts** Dil-ResNet can stably unroll KS-1D for much longer than the trajectories seen in training. This is expected, as the distribution of states is stationary in time (see Appendix). By contrast, the distribution of states in COMPDECAY-3D is not stationary in time, as the flow decays progressively into smaller and smaller structures. We can use this to construct a more challenging generalization setting by training Dil-ResNet on early stages of the decay only ($t < 1$), and testing on longer durations in which the turbulence develops further than observed during training. ($1 < t < 4$ in Fig. 2b, d). We find Dil-ResNet remains competitive with $32^3$ `Athena++` for turbulence that has decayed for up to twice longer $t < 2$ than observed during training.

**Generalization to different initial conditions** We varied the ratio of solenoidal to compressive components in the initial velocity field in COMPDECAY-3D (Fig. 5a) in order to test the models' ability to generalize to novel initial conditions. Compared to `Athena++` at $32^3$ and $64^3$, we found that the models generalize well to more solenoidal but not to more compressive initial conditions, possibly due to faster turbulence decay under compressive conditions[4]. We find that Dil-ResNet (with or without noise) has particular difficulty generalizing to the more compressive components. This is somewhat ameliorated with the added constraint to the loss function (Con-Dilated-ResNet). Results for other models are similar (see Appendix).

**Generalization to larger boxes** We tested the generalization capability of Dil-ResNet to COOLMIXLAYER-3D boxes with different size $L$ in the $x$ and $y$ direction (the length along the vertical $z$ axis perpendicular to the mixing layer remained constant). We quantified the predicted cooling velocity (small laminar flow in the low-density phase perpendicular to the interface, see Appendix for more details) which is of scientific relevance and known to depend on the box size. Specifically, for COOLMIXLAYER-3D, the dynamics undergo a transition when the box width increases relative to the cooling length,which is difficult to study because simulations for these widths essentially prohibitive. We found that unless Dil-ResNet is trained on a range of sizes, the predicted cooling velocity does not follow the right trend, and even when trained on a range of sizes, generalization beyond the training

---

[4]Note the baseline error is expected to decrease for more compressive components due to faster decay.

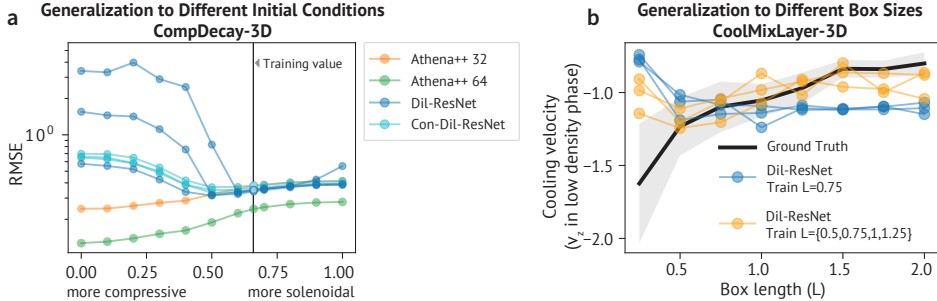

Figure 5: Generalization outside of the training distribution. (a) Generalization to different initial conditions. We vary the ratio of compressive and solenoidal components in the initial velocity field. Solid markers indicated the training region. We find that, compared to coarse `Athena++`, generalization to more compressive initial states is challenging for learned models and inconsistent across seeds, but that loss constraints (Con-Dil-ResNet) ameliorate this to some extent. (b) Cooling velocity generalization as function of the box size in the mixing layer. The black line indicates the ground truth cooling velocity averaged across time for 8 test trajectories with different box sizes, with one standard deviation represented as the shaded region. None of 3 seeds of Dil-ResNet trained on a single length of $L = 0.75$ (blue) generalize outside the training range. Dil-ResNet trained on multiple lengths (orange) shows better generalization performance. However, generalization far beyond training box sizes remains a challenge.

range is not reliable across seeds (Fig. 5b). We speculate that achieving this form of generalization would require either stronger inductive biases, more sophisticated dataset engineering, or both.

## 6    CONCLUSIONS

Learned simulation techniques have been advancing steadily in recent years. Our results demonstrate that learned simulators can outperform comparably coarse solvers in challenging turbulence settings on a range of scientifically relevant statistical measures, suggesting that state-of-the-art learned simulators can be useful for efficient, accurate simulation even in chaotic domains. Learned simulators can perform particularly well in terms of preserving high frequency information. The Dilated ResNet model had lower log Energy Spectrum RMSE than both $32^3$ `Athena++` and the higher resolutions $64^3$ `Athena++`, and the RMSE remained lower even beyond the duration seen during training. In our experiments, we found that using training noise and temporal downsampling improved the stability and accuracy of extended rollouts.

Out-of-distribution generalization is a known challenge for learned simulation (Duraisamy et al., 2019) and for deep learning methods generally, and we find that the coarse `Athena++` models outperform the learned models on more extreme generalization conditions. However, certain training choices can make learned models more robust. In particular, physically-informed regularization (e.g., Con-Dil-ResNet and Con-TF-Net) improved generalization to different initial conditions on COMPDECAY-3D, and for COOLMIXLAYER-3D, augmenting the training data with a larger range of box sizes improved generalization performance on an even larger range. Future work should explore methods from other areas of deep learning for improving adversarial robustness and domain generalization, such as feature denoising (Xie et al., 2019) and discriminative objectives (Tzeng et al., 2017; Lample et al., 2017).

More broadly, we conclude that a key potential role for learned simulators in the near term is distilling expensive, high-resolution engineered simulators into low-resolution algorithms to strike better performance-versus-efficiency tradeoffs. Our approach can be applied effectively to diverse and challenging environments, which suggests that even generic CNN-based models may apply well to low-resolution simulation on a wide range of physical domains which can be represented as grids.

ACKNOWLEDGMENTS

We would like to thank Meire Fortunato, Matthew Grimes, and Lyuba Chumakova for helpful discussions and comments on the manuscript.

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

# A    APPENDIX

| Hyperparameter | Dil-ResNet | U-Net | TF-Net | FNO |
|---|---|---|---|---|
| Kernel size | 3 | 3 | 3 | |
| Latent size | 48 | | | 48 |
| Activation | ReLU | ReLU | ReLU | ReLU |
| Loss | MSE | MSE | MSE | MSE |
| | | | | |
| Dilated block depth | 7 | | | |
| Dilated block dilations | (1, 2, 4, 8, 4, 2, 1) | | | |
| # processor blocks $N$ | 4 | | | |
| Shared processors? | No | | | |
| | | | | |
| # U-Net layers | | 7 | 4 (x3 encoders) + 3 (single decoder) | |
| Stride | | 2 | 2 | |
| CNN stack depth | | 3 | 3 | |
| Base latent size | | 64 | 64 | |
| Spatial downsample by layer | | (1, 2, 4, 8, 4, 2, 1) | (1, 2, 4, 8, 4, 2, 1) | |
| Latent sizes | | (64, 128, 256, 512, 256, 128, 64) | (64x3, 128x3, 256x3, 512x3 256, 128, 64) | |
| | | | | |
| Spatial filter kernel size | | | 5 | |
| Length of input sequence $C$ 1 | 1 | 1 | 6 | 1 |
| | | | | |
| # modes | | | | 8 or all |
| # layers | | | | 4 |
| Constraint weight | 1 | | 1 | |

Table A.1: Model hyperparameters for the four basic models used. Con-Dil-ResNet and Con-TF-Net have the same architectural parameters as Dil-ResNet and TF-Net, respectively, with the added constraint weighted by the constraint weight. Note that spatial filter kernel for TF-Net is different from the kernel size in the first row: this spatial filter is applied during the decomposition in the TF-net model.

# B    ADDITIONAL MODEL DETAILS

**Encode-U-Net-Decode (U-Net)** We designed this architecture as a simplification of the TF-Net baseline explained below. It uses a simple linear CNN encoder and decoder (same as Dil-ResNet) and a standard U-Net block (Ronneberger et al., 2015). The standard U-Net block consists of 7 layers. The first 3 layers each consist of a CNN stack followed by a factor-of-2 downsampling via striding the input. The fourth layer of the U-Net is another CNN stack, which at this point operates over a grid that is 8 times smaller than the input along each spatial axis. The last 3 layers each consist of a factor-of-2 upsampling using nearest neighbors, concatenation with the output of one of the first three layers (in reverse), and another CNN stack. This downsampling/upsampling process allows long-range communication similar to the Dilated ResNet, while also preserving local structure. As in (Ronneberger et al., 2015), each CNN stack is made of 3 CNNs, followed by activations. Other hyperparameters of the U-Net also matched those used in (Ronneberger et al., 2015).

**Turbulent Flow Net (TF-Net) (Wang et al., 2020)** TF-Net uses a domain-specific variation of U-Nets to model turbulence, along with other architectural domain-specific elements inspired by RANS-LES Hybrid Coupling in the encoder. The model implements a custom encoder by taking a sequence of the most recent $C = 6$ states as inputs, and uses this to build three separate input arrays, by applying specific combinations of learned spatial filters, learned temporal filters, and differences

of those. Each of these components are processed by a separate U-Net encoder (first 4 layers of the U-Net), then the latent outputs of each layer are concatenated and fed into a shared U-Net decoder (final 3 layers). Hyperparameters also matched those used in (Ronneberger et al., 2015; Wang et al., 2020).

**Constrained Turbulent Flow Net (Con-TF-Net) (Wang et al., 2020)** Wang et al. also explored a constraint term in the loss which penalizes the divergence, which was constrained to be zero everywhere for the PDE in their incompressible turbulence experiments. We implemented a similar model, adapting the constrained function to match either constraints or preserved quantities in our datasets. For KS-1D, the mean is constrained to be 0; for INCOMP-2D, the divergence is constrained to be 0, and for COMPDECAY-3D the total energy is constrained to be the same as the total energy of the input.

**Constrained Dilated ResNet (Con-Dil-ResNet)** This model is Dil-ResNet (described above) with the additional constraint terms from (Con-TF-Net) added to the loss function. This allows us to examine the effect of the loss term separately from the other aspects of the TF-Net architecture.

**Fourier Neural Operator (FNO) (Li et al., 2020a)** This model uses the neural operator formalism introduced in earlier work (Li et al., 2020c;b), applied in the Fourier domain. This model applies matrix multiplications in the Fourier-transformed grid with complex weights learned independently for each component, as well as linear updates The model combines pixel-wise learned linear embeddings in the spatial domain The model combines learned linear layers, applied to each point of the spatial grid separately, with matrix multiplication on the Fourier-transformed grid using learned weights. A unique matrix of weights is learned for point in the Fourier grid. Low-pass filtering is optionally implemented by truncating all but $K$ modes along each dimension in the Fourier-transformed grid. The authors explored using a version of the model that takes a stack of past states as input. Since the state is fully-Markov, we opted for the version of the model that takes a single state as input as it is more comparable to the majority of the other models. Unlike previous models, which rely on spatially local convolutions and depth for long-range communication, this model applies long-range communication at every step in the Fourier domain. We implemented this model for 1D and 2D spatial domains as in (Li et al., 2020a).

We found difficulties running this model on the 3D datasets due to compute and memory constraints. We speculate this is due to time and space complexity of the Fourier transform (when taking into account the backwards pass) which would probably require a custom gradient re-materialization strategy. This may be specific to TensorFlow which required us to reimplement the non-differentiable `tf.signal.rfft3d` as a sequence of `tf.signal.rfft2d` and `tf.signal.fft`.

## B.1 Additional dataset details

### B.1.1 1D Kuramoto-Sivashinsky (KS) Equation

This is a well-studied 1D PDE that generates unstable, chaotic dynamics in 1 dimension (Kuramoto, 1978; Sivashinsky, 1977) with periodic boundaries. The ground truth simulations are computed using the Fourier Spectral Method. Initial condition was set to $\cos(w_1 x + \phi_1)(1 + \sin(w_2 x + \phi_2))$, where x ranges from 0 to $2\pi$, $\phi_i$ are sampled uniformly from $[0, 2\pi)$, and $w_1$ are integers sampled uniformly from $[1, 12]$. We perform a *warmup* from this initial condition for 75 simulation time units.

### B.1.2 2D Incompressible Decaying Turbulence

This models fluid flow described by the Navier-Stokes equations in which small-scale eddies decay into the large-scale structures due to the inverse energy cascade (Boffetta & Ecke, 2012b). The underlying simulations were performed by solving the incompressible Navier-Stokes equations using a direct numerical simulation (DNS) finite-volume solver. Boundaries along both dimensions are periodic, and the initial conditions consist of random velocity fields with small-scale variation. Initial conditions for different trajectories were obtained by sampling a high-resolution velocity field from a log-normal distribution of amplitude 1 and wavenumber $k = 10$. We perform a *warmup* for 500 simulation time units, this is done to discard the transient flow that is heavily influenced by the underlying numerical scheme.

| | KS Equation | Incompressible Decaying | Compressible Decaying | Compressible Radiative Cooling Mixing Layer |
|---|---|---|---|---|
| Numerical Solver | Fourier Method | DNS | `Athena++` | `Athena++` |
| # Spatial dims | 1 | 2 | 3 | 3 |
| # Features | 1 | 2 | 5 | 5 |
| Features | $v$ | $v_x, v_y$ | $\rho, v_x, v_y, v_z, P$ | $\rho, v_x, v_y, v_z, P$ |
| | | | | |
| Box size | | | | |
| $L_x$ | $2\pi$ | $2\pi$ | 1 | 0.25 to 2 |
| $L_y$ | n/a | $2\pi$ | 1 | 0.25 to 2 |
| $L_z$ | n/a | n/a | 1 | 3 |
| | | | | |
| Grid element size | | | | |
| *Solver* | $2\pi$ / 256 | $2\pi$ / 576 | 1 / 128 | 1 / 128 |
| *Learned model* | $2\pi$ / 64 | $2\pi$ / 48 | 1 / 32 | 1 / 32 |
| *(relative to solver)* | 4x | 12x | 4x | 4x |
| | | | | |
| *Warm-up* duration | 75 | 500 | 0.05 | 2.32 |
| Trajectory duration | 181 | 400 | 1 | 7.226 |
| | | | | |
| Time step | | | | |
| *Solver* | 0.5 | 0.00436 | 0.0005 | 0.00012 to 0.00014 |
| *Learned model* | 0.5 | 3.35 | 0.032 | 0.12 |
| *(relative to solver)* | 1x | 768x | 64x | 1000x to 875x |
| | | | | |
| # Trajectories | | | | |
| *Training* | 1000 | 190 | 27 | 20 if $L_x = 0.75$ |
| | | | | 5 if $L_x \neq 0.75$ |
| *Validation* | 100 | 10 | 4 | 1 per $L_x$ |
| *Test* | 100 | 10 | 4 | 1 per $L_x$ |
| | | | | |
| Training details | | | | |
| Early stopping? | No | No | No | Yes |
| Batch size | 32 | 8 | 1 | 1 (4 if multisize) |
| Noise | 1e-2 | 1e-4 | 1e-2 | 1e-3 |
| Constrained | mean $v$ | divergence | total energy | 1e-3 |

Table B.1: Dataset details. $\rho$ refers to density, $P$ to pressure, and $v_x, v_y, v_z$ to velocity components. *Warm-up* refers to the initial transient from initial conditions which is influenced by the underlying numerical scheme and discarded from evaluation and training. Figures are dimensionless.

| # Parameters | 1D | 2D | 3D |
|---|---|---|---|
| Dil-ResNet | 195361 | 584162 | 1757477 |
| U-Net | 3916993 | 11739202 | 35072581 |
| TF-Net | 17358540 | 31239137 | 93712968 |
| FNO (K=8) | 288433 | 4159250 | 62222693 |
| FNO (K=full) | 1191601 | 42479378 | 603994469 |

Table B.2: Number of parameters per model.

### B.1.3 3D COMPRESSIBLE DECAYING TURBULENCE

This models decaying transonic Navier-Stokes turbulent flow in a 3D cubic box with periodic boundary conditions (e.g., Orszag & Patterson, 1972). These simulations adopt an adiabatic equation of state with a constant adiabatic index $\gamma = 5/3$. Simulations were carried out with `Athena++` (Stone et al., 2020). The initial turbulence is driven on scales $\geq L$, the size of the box. The turbulent

driving pattern in the initial condition is split into its compressive and solenoidal components using a Helmholtz decomposition. The relative strength of these two components is varied from purely compressive to purely solenoidal. The initial driving pattern results in a root-mean-squared velocity of $\sqrt{2}c_s$, where $c_s^2 = (5/3)(P/\rho)$ is the sound speed of the fluid. The initial conditions are varied across trajectories by randomizing the phase of the spectral components, leading to different pattern in real space. We perform a *warmup* for 0.05 simulation time units. Compressible turbulence is ubiquitous across problems in science: all fluids are somewhat compressible, and this is usually non-negligible when modeling gases (Pope, 2000). In astrophysics problems, for which the `Athena++` was specifically developed, understanding the dynamics and properties of these flows on small and large scales plays a crucial role in regulating planet, star, black hole, and galaxy formation (Brandenburg & Nordlund, 2011).

### B.1.4    3D COMPRESSIBLE RADIATIVE COOLING MIXING LAYER DYNAMICS

These simulations model the interplay of radiative cooling and mixing that results from turbulence driven by the Kelvin-Helmholtz instability, which can arise when there is velocity difference across the interface between two fluids of different densities. The simulations are set up as a boundary problem initialized with a low-density fluid ($\rho = 0.01$) on the top half of the domain ($z > 0$) moving in the positive $x$ direction ($v_x = 2.04, v_y = v_z = 0$), and a high-density fluid ($\rho = 1.$) moving in the negative x direction in the bottom half of the domain ($v_x = -2.04, v_y = v_z = 0$). The initial pressure is set to 1. The boundary conditions are periodic in both $x$ and $y$, and fixed in $z$. To break the symmetry, small perturbations to $v_z$ are added along the boundary. These perturbations are changed across trajectories by randomizing the phase of the spectral components, leading to different patterns in real space. We perform a *warmup* for 2.32 simulation time units. Simulations were carried out with `Athena++` (Stone et al., 2020).

This data presents a few unique challenges compared to the others. First, it was the only domain with fixed boundary conditions. Second, because turbulent dynamics are limited to the vicinity of the mixing layer, and because the behavior of the fluid above, at, and below the mixing layer is markedly different, there is less data representative of each type of fluid dynamic behavior.

## B.2    ADDITIONAL MODEL DETAILS

The learned simulator consisted of a CNN encoder, a dilated CNN processor with residual skip connections, and a CNN decoder. The parameters of the model are listed in Table A.1.

**CNN parameters**    All individual CNNs use a kernel size of 3 and have 48 output channels (except the decoder, which has an output channel for each feature). Each individual CNN layer in the processor is immediately followed by a rectified linear unit (ReLU) activation function. The Encoder CNN and the Decoder CNNs do not use activations.

**Dilated connections**    The dilation rate in the CNN filter introduces space between each element in the filter. Whereas a standard CNN filter with kernel 3 would operate over 3 adjacent pixels, a dilated CNN filter with kernel 3 and dilation 2 will operate over 3 pixels spaced at intervals of 2 in each dimension, spanning a field 5 pixels wide. This increases the scale of the CNN kernels while preserving the number of parameters and the resolution.

**CNN padding**    For each dimension in $\mathcal{X}$ with periodic boundary conditions, we implemented periodic padding. For dimensions with a fixed boundary condition, we forced the boundary to a value rather than letting the model predict it, and masked the loss so the model was not trained to predict boundary conditions. The tensor was padded with repetitions of the boundary value. We also augmented the input state with a feature that indicated fixed-boundary versus non-boundary states using a one-hot vector, so the model could distinguish them.

## B.3    ADDITIONAL TRAINING DETAILS

**Training noise**    In some cases, we trained with Gaussian random noise with fixed variance $\sigma$ added to the input $X_t$ of the loss function. Note that this not only affects the input to the neural network, but

also slightly modifies the target $\Delta X = X_{t+\Delta t} - X_t$ (and also impacts the variance used to normalize targets).

**Loss**    At training time we sample pairs of input-output states (separated by the model time step) from the trajectories, and perform gradient updates based on a single step of the model. We do not rollout the model during training.

**Optimization**    We optimized the loss using an Adam optimizer. We trained the models for up to 10M steps, with exponential learning rate decay annealed from $1e - 4$ to $1e - 7$ in the first 6M steps. Models usually reached convergence at around 5M steps. Training took up to a week on an NVIDIA V100 GPU.

**Hyper-parameter optimization**    We did not perform exhaustive hyper-parameter optimization, except on the scale of the noise (which we scanned for each domain) and some informal tuning of the depth of the dilated blocks and latent size (to make sure the network had enough capacity for the most challenging domain). Note that achieving optimal performance on test trajectories from the training distribution was not part of the scope of this work, and there are likely hyperparameters that would improve performance over our results.

**Validation**    All of the research was performed by looking at performance on the validation sets. The test set was completely held-out until final evaluation prior to writing the paper.

**Early stopping**    For the Mixing Layer Turbulence (COOLMIXLAYER-3D), which was more prone to overfitting, we used early stopping based on the validation performance. For all other models we simply evaluated the model as it was at the end of training.

## B.4    ADDITIONAL RESULTS

Three instances of each model were trained 3 times using 3 different initialization seeds. Bar plots indicate median performance and bars indicate min-max performance. We chose the energy field $E = \frac{1}{2}\rho v^2 + \frac{3}{2}P$ as the quantitative metric of 3D turbulence for the main text, as it summarizes performance on all state variables.

Figures and videos for 3D environments (COOLMIXLAYER-3D and COMPDECAY-3D) show a single slice of the 3D grid at $y = 0$, displaying the $x$ and $z$ coordinates in the horizontal and vertical axis, respectively.

**Downsampling in time**    Fig. B.6 as well as Videos show models for KS-1D, INCOMP-2D, and COMPDECAY-3D models trained and working well on a wide range of time step sizes. Note that for a fair comparison in Fig. 3d,f performance is averaged across only time steps that are predicted for all models (e.g. multiples of the largest time step, 256).

**Downsampling in space**    All comparisons across spatial resolutions are always obtained by first downsampling the data into a common $32^3$ grid, using an approach that preserves mass, momentum, and energy (Same approach used by Athena++). Fig. B.2 (top row) shows the downsampling comparisons equivalent to Fig. 2g-i for each of the state variables independently.

**Running time**    For 1 simulation time unit of COMPDECAY-3D, the CPU runtime for Athena++ with 8 CPU processors is $\sim$4s, $\sim$60s, and $\sim$1000s for $32^3$, $64^3$, and $128^3$ resolutions, respectively (quartic scaling, as the time step is also scaled with the resolution to compensate for numerical viscosity). In comparison, the learned model's runtime is 1s on an NVIDIA V100 GPU, and 20-30s on a 8-core CPU. Note that the learned model runs at reduced spatial and temporal resolution, but preserves the dynamics of the high resolution $128^3$ Athena++ simulation. Simulations in Athena++ may be faster if implemented for GPUs. However, because scientific simulators like Athena++ are specialized, each simulator's GPU implementation requires specialized engineering effort, whereas learned models can take advantage of methods designed more generally for deep learning.

**Cooling velocity**    Having reliable models that generalize to larger boxes increases the applicability of learned models to scientific domains by enabling experiments in regions of parameter space that would otherwise be prohibitively expensive to simulate. For example, for COOLMIXLAYER-3D, the dynamics undergo a transition when the box width increases relative to the cooling length ($v_{\text{turb}}t_{\text{cool}}$), which is difficult to study because simulations for these widths essentially prohibitive. Thus, we want to understand what affects the learned simulator's ability to generalize to a range of box widths (the length $L_x$ and width $L_y$ of the input tensor $X$) not previously seen in the training data. For COOLMIXLAYER-3D, we can look at the cooling velocity, the average inflowing velocity at the low density fluid boundary that develops as a means to resupply the energy that has been radiated away in the mixing layer. Cooling velocity is a useful metric because it is a scalar quantity that depends on the box width. Furthermore, understanding how turbulent dynamics at a mixing layer pull heat from the surroundings is scientifically relevant for questions in astrophysics (Fielding et al., 2020). We evaluate generalization to different box sizes ($L_x = L_y \in [0.25, 0.5, 0.75, 1.0, 1.25, 1.5, 1.75, 2.0]$) (Fig. 5b). We find that the model trained on box length $L_x = L_y = 0.75$ (orange) is not able to produce trajectories with the correct cooling velocity for other box sizes. However, we also find that augmenting the dataset with data from a range of box sizes ($L_x = L_y \in [0.5, 0.75, 1.0, 1.25]$) improves accuracy of the cooling velocity estimate for the unseen box lengths, although the generalization outside the training domain remains imperfect and unreliable across seeds. We speculate that achieving this form of generalization would require stronger inductive biases and/or more sophisticated dataset engineering.

**Other Statistical Metrics**    The choice of metrics can impact the utility of learned simulators for scientific applications. Here we describe a few physically motivated metrics we use to evaluate the learned models.

We use the power spectral density, which we quantify for the total energy in Figure 2, to quantify how well the different models are preserving information at different resolutions. Simulators at coarse resolution are known to lose high-frequency information, and we are specifically interested understanding how learned and physics-based simulators compare in terms of preserving detailed, high-frequency components. To quantify the power spectral density, we $n$-D Fourier transform, take the magnitude at each point of the Fourier transformed signal to get the $n$-D PSD, and convert this to a 1D spectrum by taking the mean magnitude over points in the $n$-D frequency grid with the same frequency vector length to get a 1D power spectral density $\text{PSD}(k)$ over a scalar frequency $k = \sqrt{\sum k_{x_i}^2}$ for spatial dimensions $x_i$ . Frequencies with a vector of length $> 1$ (the length of the environment along 1 dimension) are cutoff, leaving only frequencies that fall within the sphere of radius 1. We plot $\log \text{PSD}$ and compute spectrum error over these frequencies. The spectra for different state variables are shown in Figure B.1 for the coarsened `Athena++` models and Dil-ResNet, and in Figure B.3 for the learned models. The error over time for the spectra is shown in Figures B.2 and B.2.

The probability distribution functions (PDFs) of the fluid quantities provides an important basis for understanding turbulent flows. Due to the inherently chaotic nature of turbulence, statistical measures are often the most robust measures of their properties. The shape and widths of PDFs are common metrics for characterizing turbulent flows(Federrath & Klessen, 2012). Moreover, joint PDFs of fluid quantities (e.g., velocity-temperature, or pressure-entropy) are used to understand more complex flow behavior and to characterize which flow components are responsible for observed behavior (Fielding et al., 2020). Joint distribution functions are often especially useful when certain physical processes only take place in specific region of phase space (e.g., combustion or radiative cooling). PDFs for different state variables and joint distributions (entropy v. pressure, density v. pressure) are shown in Figure B.1 for the coarsened `Athena++` models and Dil-ResNet, and in Figure B.3 for the learned models. The error over time for these statistics is shown in Figures B.2 and B.2.

Another measure of interest are spatial autocorrelation measures, particularly velocity autocorrelation. Like the histograms, these capture statistical structure and provide a metric that is more invariant to the chaotic behavior of turbulence. The $p^{\text{th}}$ order autocorrelation is closely related to the velocity structure function, $S_p(\ell) = \mathbb{E}_{\mathbf{r}, \mathbf{u}, \|u\|=1}[\|\mathbf{v}_{\mathbf{r}+\ell\mathbf{u}} - \mathbf{v}_{\mathbf{r}}\|^p]$, for velocity $\mathbf{v}$, defined at a position vector $\mathbf{r}$ (e.g., $\mathbf{v}_{\mathbf{r}}$), with unit vector $\mathbf{u}$ multiplied by a scalar $\ell$. Seminal self-similarity arguments that underpin turbulent theory lead to the prediction that $S_p(\ell) \propto \ell^{p/3}$ (Kolmogorov, 1941; Frisch, 1995). Structure and autocorrelation functions, therefore, provide important measures on the ability of learned simulators to capture subtle, but scientifically essential properties of turbulent flows. First-

and second-order autocorrelation functions for different state variables are shown in Figure B.1 for the coarsened `Athena++` models and the Dil-ResNet model and Figure B.3 for all models trained on COMPDECAY-3D. The error for each of these functions over time are shown in Figures B.2 and B.4.

**Model comparison**  Fig. B.7 shows predicted rollouts for KS-1D of different learned models presented in Figure 4a. Fig. B.3 shows statistics of the final frame of the predicted rollouts for different learned models presented in Figure 4. The error of each of these statistics over time is shown in Fig. B.4. Videos show the rolled out predictions for different models for INCOMP-2D and COMPDECAY-3D.

**KS-1D generalization**  Figs. B.8 and B.9 show generalization to larger domains and longer trajectories. While the learned simulations do not perfectly capture the ground truth, we see that qualitative features of turbulence are preserved across the rollout.

**COMPDECAY-3D generalization**  Figs. B.8 and B.9 show quantitative generalization performance of the Dil-ResNet model to larger domains and longer trajectories on KS-1D. While the learned simulations do not perfectly capture the ground truth, we see that qualitative features of turbulence are preserved across the rollout. Figures B.2 and B.4 show different metrics for different variables across rollout (pink background indicates the region not seen during training). Videos show the predicted rollouts when generalizing to different initial conditions and to longer rollouts.

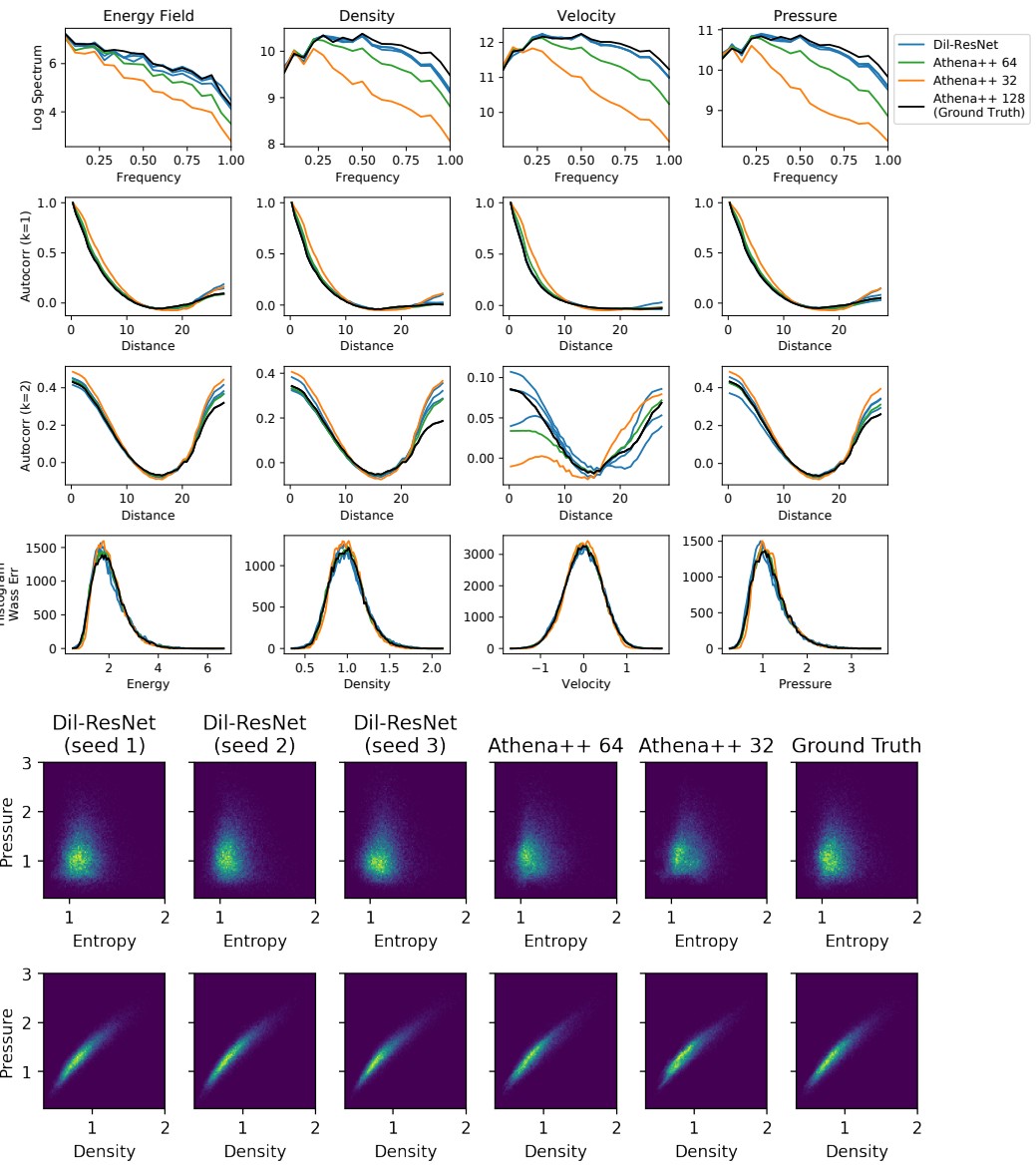

Figure B.1: COMPDECAY-3D Comparison of learned Dil-ResNet model trained with noise=0.01 to Athena++ run at resolutions of $32^3$, $64^3$, and $128^3$ (ground truth) for a variety of different metrics at $t = 0.99$ (the end of a rollout that lasts the duration seen during training).

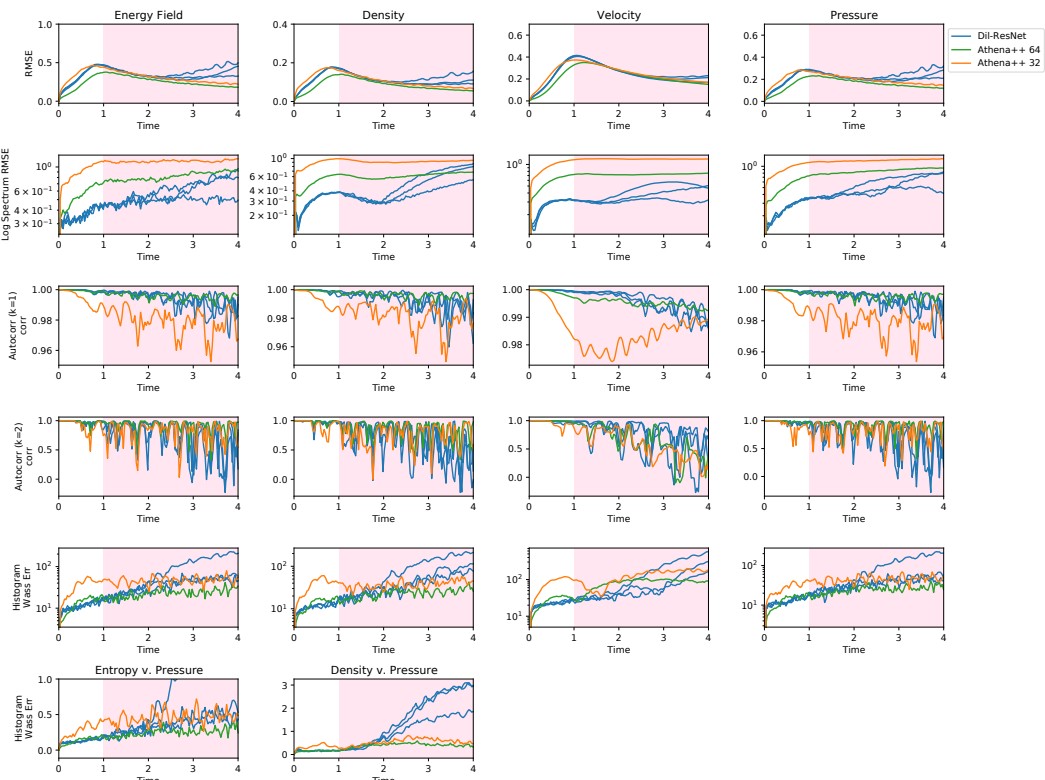

Figure B.2: COMPDECAY-3D Model metrics over time across learned Dil-ResNet model trained with noise=0.01 and Athena++ coarsened to different levels of resolution. Except for the last row, which shows the error in the Entropy v. Pressure and Density v. Pressure 2D histograms, each column is a state variable (Energy Field, Density, Velocity Components, and Pressure) and each row is the error for a different function over the data. Note rollouts start at $t = 0.16$, after 5 model steps would have elapsed, because TF-Net takes in 6 frames and all models should start at the same point. Same color scaling is used across each 2D histogram row. Background is pink for times not shown during training.

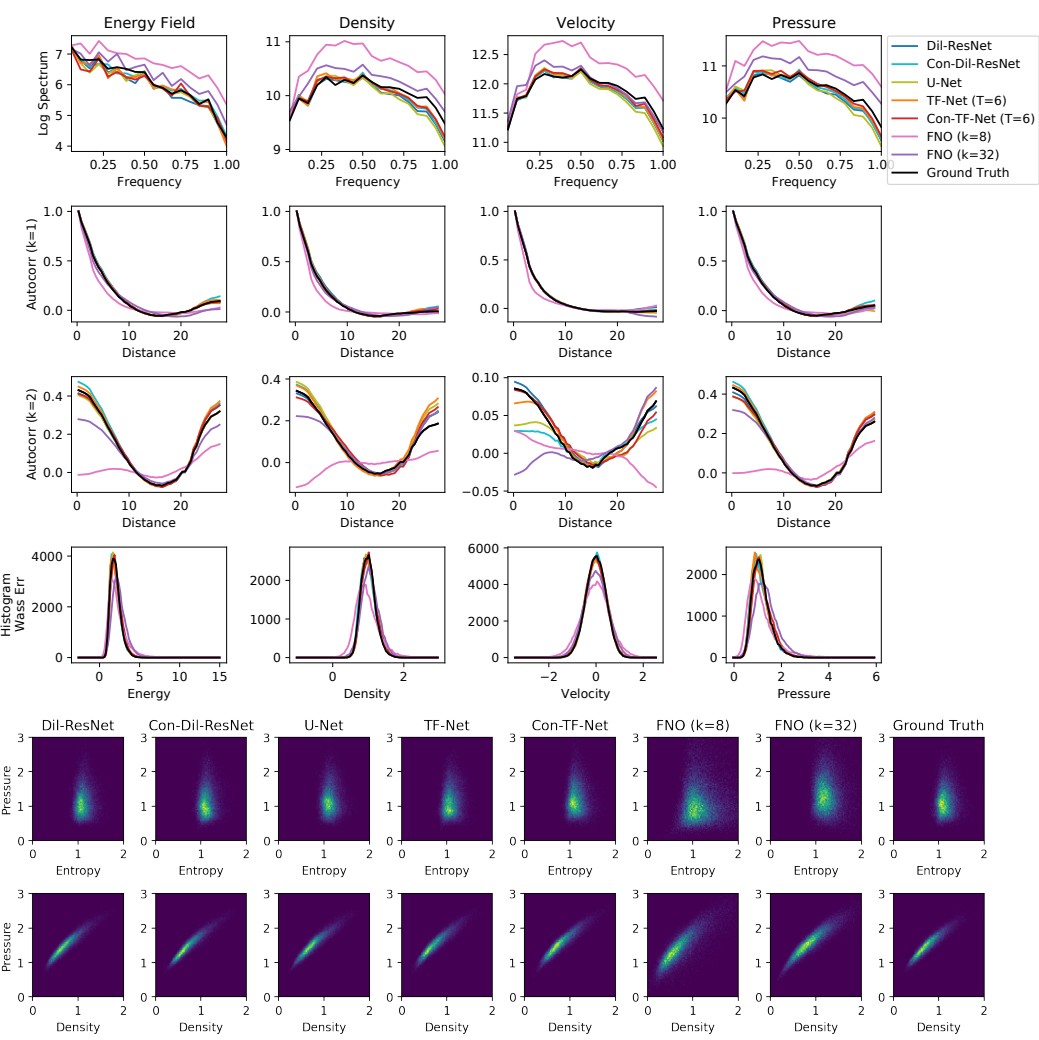

Figure B.3: COMPDECAY-3D Comparison of learned models trained with noise=0.01 for a variety of different metrics at $t = 0.99$ (the end of a rollout that lasts the duration seen during training).

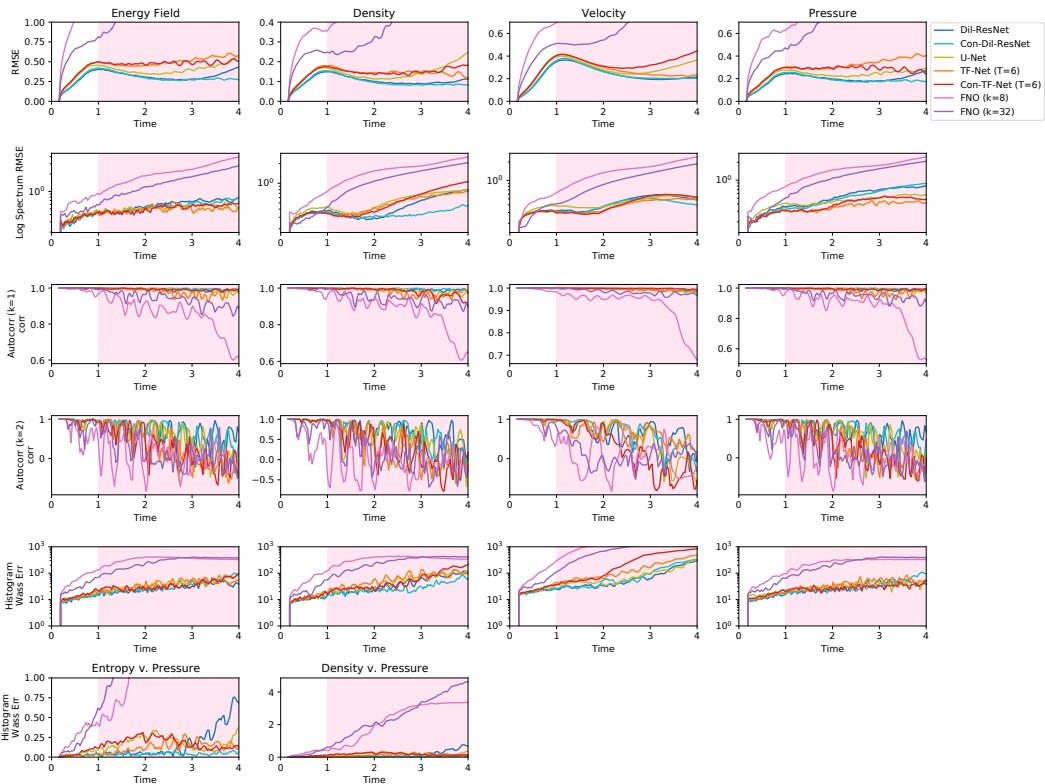

Figure B.4: COMPDECAY-3D Model metrics over time across different learned models trained with noise=0.01. Except for the last row, which shows the error in the Entropy v. Pressure and Density v. Pressure 2D histograms, each column is a state variable (Energy Field, Density, Velocity Components, and Pressure) and each row is the error for a different function over the data. Background is pink for times not shown during training. One seed shown per model (quantitative results for multiple seeds shown in Figure B.11)

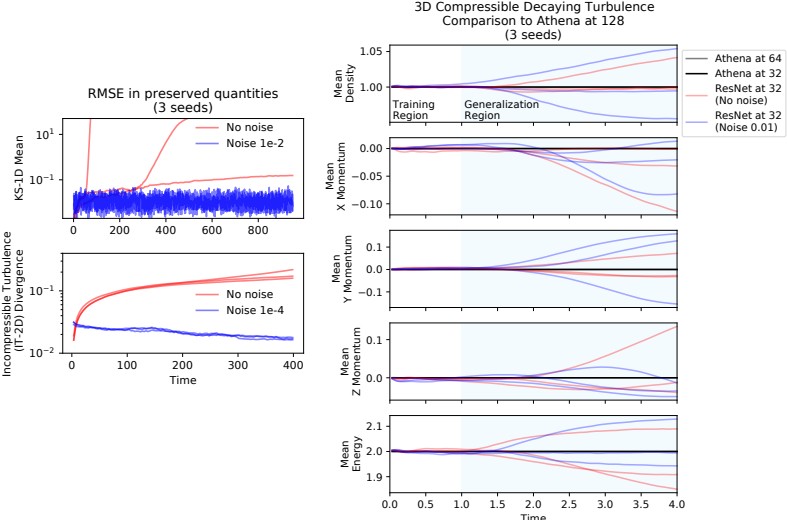

Figure B.5: (left) KS equation (KS-1D) net velocity error, and Incompressible Turbulence (INCOMP-2D) divergence error as function of model step. Training with noise helps keeping the values bounded to be close to 0. (right) Preservation of the 5 conserved quantities in 3D Compressible Turbulence (COMPDECAY-3D) as a function of time. In this case, training with noise does not completely prevent drift of the conserved quantities.

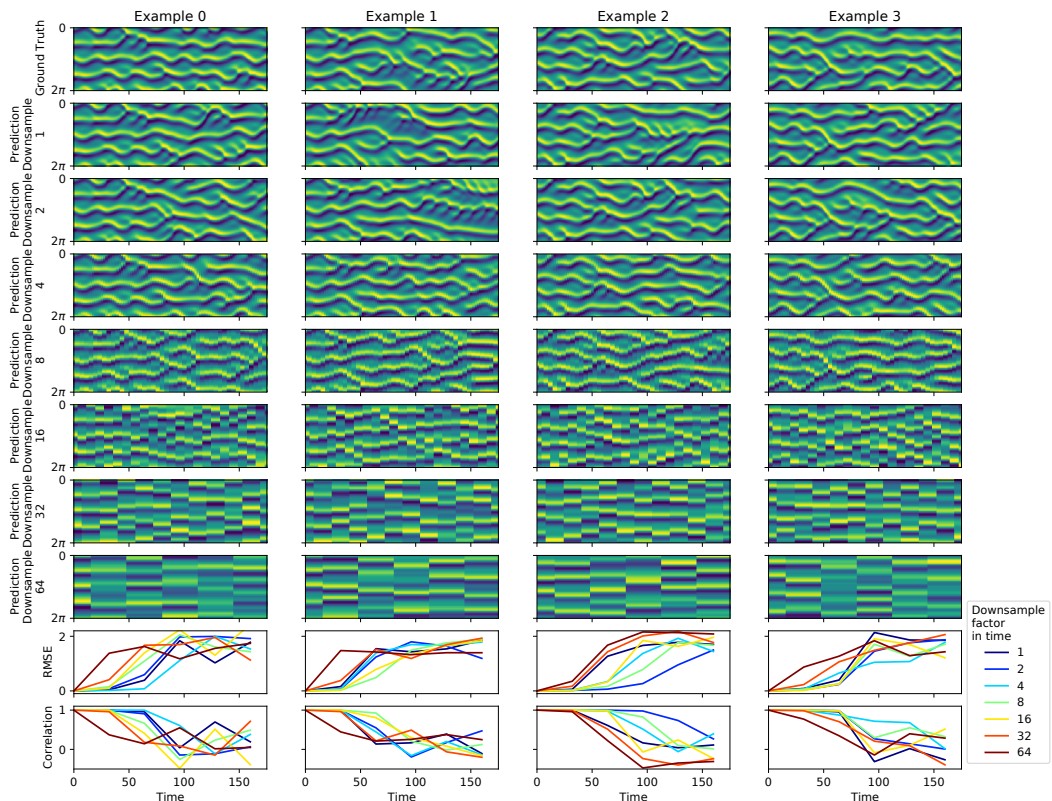

Figure B.6: (top) Sample trajectories from the model trained on the KS-1D dataset at different temporal downsampling factors. (bottom) MSE and correlation performance of ML models for different downsampling factors in time as a function of simulation steps.

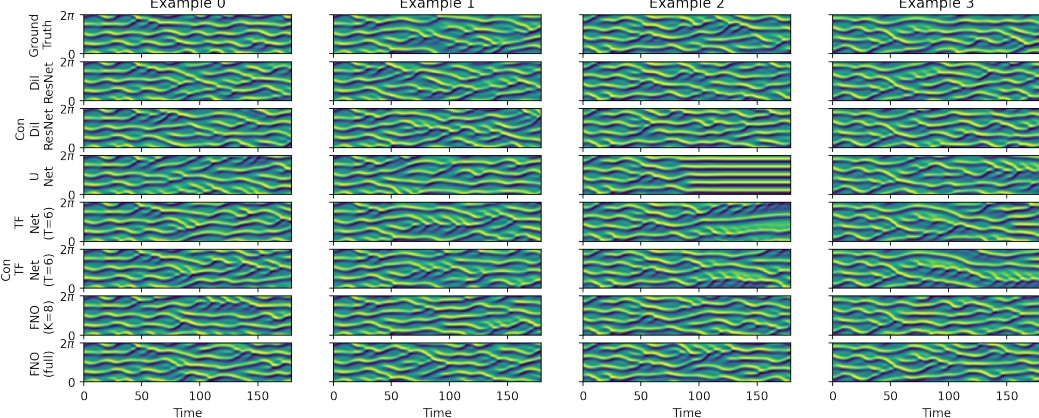

Figure B.7: KS-1D Rollouts for models trained with noise=0.01.



Figure B.8: Dil-ResNet model generalizing to larger spatial domains on KS-1D (trained on domains $2\pi$ wide).

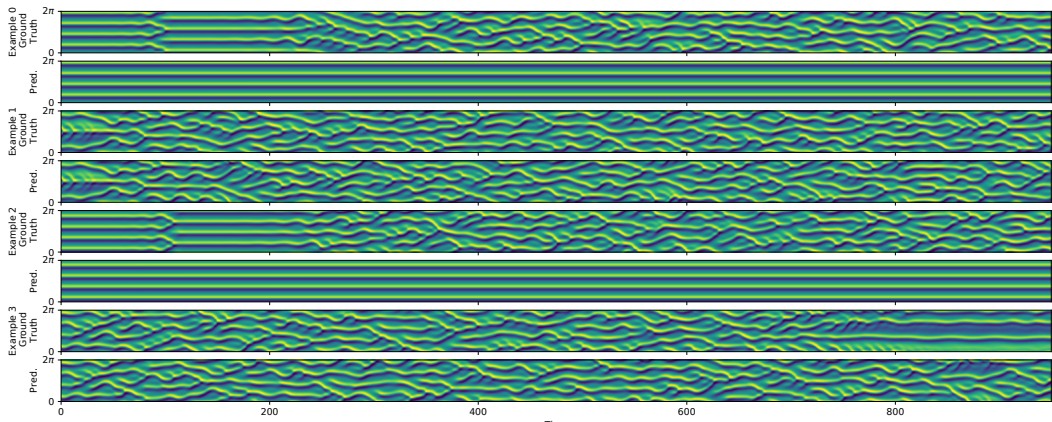

Figure B.9: Dil-ResNet model generalizing to longer trajectories on KS-1D (trained on trajectories with 181 simulation time units).

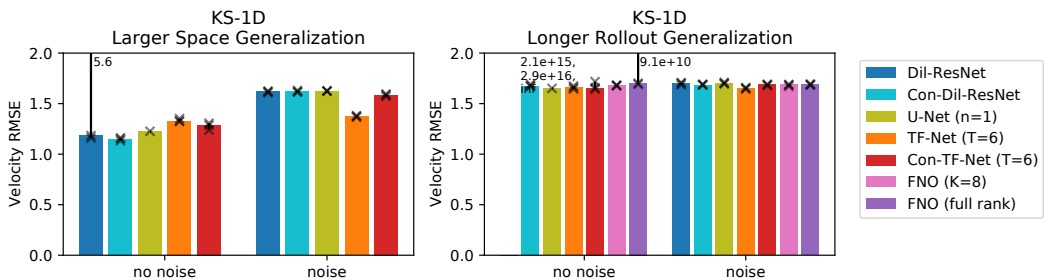

Figure B.10: Model comparison for KS-1D generalization to larger spatial domains (left) and longer rollouts (right).

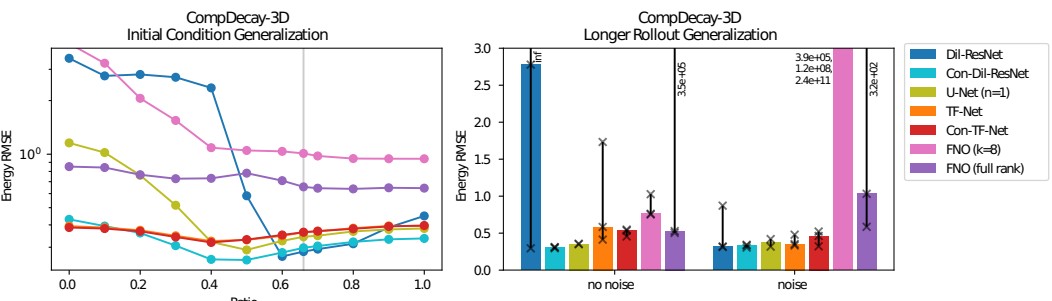

Figure B.11: Model Comparison for Generalization. (left) Generalization to different ratios of solenoidal to compressive components in the initial conditions for learned models (trained with noise=0.01). 0=more compressive, 1=more solenoidal components in initial conditions. The gray line marks the value the models were trained on (0.66). (right) Generalization to longer rollouts. Added noise during training improves stability and therefore leads to lower error on longer rollouts. Cutoff point for TF-Net with no noise has RMSE 2.99, and cutoff bar for Dil-ResNet goes to Inf (numerical error). As shown in Figure 5, we see that adding the constraint to the loss function makes the model generalize more reliably to the more compressive regime, and that training noise promotes stability for longer rollouts.. Note that these results are numerically different from Figure 5 because rollouts were started from the sixth timestep to allow comparison with TF-Net, which takes as input a sequence of length $C = 6$.

