# OpenReview forum: "Learned Simulators for Turbulence"
_ICLR.cc/2022/Conference — ICLR 2022 Poster_

### Official Review · Reviewer_YTSP · 2021-10-28

**Correctness:** 3
**Technical Novelty And Significance:** 3
**Empirical Novelty And Significance:** 3
**Recommendation:** 8
**Confidence:** 4

**Main Review:**

The authors compare a variety of models to the proposed dResNet, with a simpler Unet, the TF net, and a Fourier-Op network. It was, e.g., interesting to see that the latter does fairly badly in some of the tests. They also evaluate how introducing noise in the predictions helps to stabilize the rollouts.

The results contain a nice variety, and it is impressive to see predictions of full 3D turbulence simulations. The resolutions are not huge, with 32^3, but nonetheless contain complex dynamics.

The paper also contains an interesting evaluation of generalization for different initial conditions and simulation domain sizes. The longer rollouts are also an important point to study in this context. Here I found the presentation of the paper unclear: how many steps did the NN actually perform? The captions often give conflicting information: e.g., in Fig. 2, the time axis goes from 0 to 4, and the caption talks about t=992. It would be important to specify these durations more clearly, such that readers know how many steps the models predicted independently.

Minor, but for divergence constraints I'd recommend to cite the Tompson et al. 2018 paper "Accelerating eulerian fluid simulation with convolutional networks". Also, the title of the website and on OpenReview differ. "Efficient turbulence" sounds very generic to me, being more specific in the title is a good idea.

Here are a few additional points which I hope the authors can clarify in the rebuttal:

- For the proposed dResNet architecture, what is the role of the encoder decoder? If it's just a single linear layer, why is it so important? Are there comparisons showing the usefulness of the encoding-decoding step?

- The graphs in Fig. 3 (and others) are confusing - time is shown as 0 to 1, but the caption claims time steps of \Delta t = 1, and on the right up to 128. What are the real sizes, and how many time steps were evaluated?

- In Table A1, no model sizes are given. How many parameters did the different models have?


**Summary Of The Paper:**

This paper proposes a method to predict turbulent dynamics of coarse simulations. To achieve this, a modified ResNet architecture is proposed, which the authors call Dilated ResNet (dResNet). This dResNet is combined with an encoder/decoder model inspired by graph networks. Unfortunately, the details of this architecture are not made very clear, but I hope this is something the authors can adress in future revisions of their work. Beyond the proposed architecture, the paper presents an interesting range of test cases from 1D KS to 3D compressible turbulence.


**Summary Of The Review:**

In summary, the paper represents an interesting read, and contains a variety of very interesting results. There are a few open points, but, giving the authors the benefit of doubt, I hope that these can be cleared up during the rebuttal. Given my current understanding of the work I would argue for accepting the paper to ICLR.

---

> ### Author Response · Authors · 2021-11-19
> **Response to Reviewer YTSP**
>
> Thank you for your thoughtful review. We answered your questions below.
>
> > The resolutions are not huge, with 32^3, but nonetheless contain complex dynamics.
>
> We just want to make it clear that the resolution of the learned simulator is 32^3, but the training data is produced by running a ground truth simulator at 128^3 and then downsampling these runs to 32^3. This is what allows the dynamics to be so complex and the results of the learned model to be better than the numerical solver ran at lower resolutions.
>
> > how many steps did the NN actually perform?
>
> The number of steps performed by the NN depends on $\Delta t$. For Fig 2, $\Delta t$ = 32e-3 , as this is the value that minimizes rollout loss (Fig 3). Thus, the number of model steps to as close to t=1 as possible without going over is 31. For the other values of $\Delta t$ shown in Fig 3, the number of rollout steps varies, but the length of the rollout is always fixed at $t\approx1$. We have now clarified this in the updated version.
>
> > The captions often give conflicting information: e.g., in Fig. 2, the time axis goes from 0 to 4, and the caption talks about t=992.
>
> Thank you for pointing this out – the caption should be $t=0.992$ (end of the training window, $t\approx1$, as close to $t=1$ without going over), and Fig 3 $\Delta t$ is in units of 1e-3. These have been updated in the manuscript to $t\approx1$.
>
> > Minor, but for divergence constraints I'd recommend to cite the Tompson et al. 2018 paper "Accelerating eulerian fluid simulation with convolutional networks".
>
> Thank you for the suggestion, we have added this citation to the Related Work.
>
> > Also, the title of the website and on OpenReview differ. "Efficient turbulence" sounds very generic to me, being more specific in the title is a good idea.
>
> Thank you for pointing this out. We will make sure the OpenReview title to match the pdf, and use the more specific title “Learned Coarse Models for Efficient Turbulence Simulation” to communicate that the efficiency boost comes from coarsening.
>
> > For the proposed dResNet architecture, what is the role of the encoder decoder? If it's just a single linear layer, why is it so important? Are there comparisons showing the usefulness of the encoding-decoding step?
>
> The role of the encoder is to match the dimensionality of the input features to the latent size, and the role of the decoder is to match the dimensionality of the latent size to the output size. We will make this more explicit in the text.
>
> To elaborate a bit more, the reason this is important is because it enables adding residual connections (which from research in vision models are key to avoid vanishing gradients and to training deeper models) around most of the learnable deep layers of the model. Specifically, residual connections require layers with identical input and output sizes (so input and output can be added together), so the sooner the model starts operating on a common latent size as input and output size for the layers, the better. For this purpose, we chose simple linear layers as a simple way to quickly match the input/output dimensionality to the latent size.
>
> > The graphs in Fig. 3 (and others) are confusing - time is shown as 0 to 1, but the caption claims time steps of \Delta t = 1, and on the right up to 128. What are the real sizes, and how many time steps were evaluated?
>
> Thanks, as mentioned above, this was an error: we accidentally used milli-time units, instead of regular time units– $\Delta t$ ranges from 1e-3 to 128e-3 time units. We have updated the manuscript to remove the confusing usage of milli-time units.
>
> > In Table A1, no model sizes are given. How many parameters did the different models have?
>
> We will add a table with the number of parameters to the appendix, also including it here for convenience.
>
> \# params per model |          1D  |           2D   |          3D
> -------------------|--------------|----------------|-----------
> Dil-ResNet         |      195361  |       584162   |     1757477
> U-Net              |     3916993  |     11739202   |    35072581
> TF-Net             |    17358540  |     31239137   |    93712968
> FNO (K=8)          |      481608  |      7729184   |        -
> FNO (K=full)       |     1926433  |     69562658   |        -

---

> > ### Comment · Reviewer_YTSP · 2021-11-23
> > **Post rebuttal**
> >
> > I'd like to thank the authors for their clarifying comments.
> >
> > I have already initially supported acceptance of this submission, and the authors have clarified the open points of my review. While I can understand some of the concerns voiced in the other reviews, I still think this is an interesting submission that would make for a very nice ICLR paper. So I believe an "accept" rating is very appropriate.

---

### Official Review · Reviewer_HsnX · 2021-11-02

**Correctness:** 3
**Technical Novelty And Significance:** 2
**Empirical Novelty And Significance:** Not applicable
**Recommendation:** 6
**Confidence:** 3

**Main Review:**

The demonstration doesn’t clearly show the benefits of this method. What’s the difference between the three blue lines in figs 2b and 2d? Where is the red line that is supposed to denote the ground truth? Isn’t the error is larger in Dil-ResNet in fig 2b? What’s the point of t=1000 for a simulation that is already imprecise at t=1? Is it t=1 or t=1000 in figs 2f-2i? There are similar issues in other figs, like figs 4 and 5. It’s hard to show significant advantages with these low-accuracy simulations.

As I said before, it would be more convincing if the authors can demonstrate some standard examples, such as Taylor Vortex, Karman Vortex, Frog-leaping, etc. For some turbulent flows that are chaotic themselves, the solutions at the resolution of 32 are meaningless, and the results are even worse than some large-eddy simulations.


**Summary Of The Paper:**

This paper present a  neural network-based model by training learned simulators at low spatial and temporal resolutions to capture turbulent dynamics generated at high resolution. It also shows that the proposed model can simulate turbulent dynamics more accurately than classical numerical solvers at the same low resolutions across various scientifically relevant metrics. Finally, a series of numerical examples including one-dimensional, two-dimensional, and three-dimensional nonlinear equations are used to demonstrate the feasibility of the algorithm. I reviewed this paper in NeurIPS 2021, which was titled “Learned Simulators for Astrophysical Turbulence”. The authors revised the paper based on previous reviews. Thanks for the efforts the authors have made.

**Summary Of The Review:**

The authors do make efforts on this paper, however, only some statistical presentation of some low-precision turbulence simulation data is not convincing enough to make it attractive.

---

> ### Author Response · Authors · 2021-11-19
> **Response to Reviewer HsnX**
>
> Thank you for your thoughtful review (as well as your comments from the last round of review, which were helpful in improving the paper and clarifying our aims). We address your questions below.
>
> > What’s the difference between the three blue lines in figs 2b and 2d?
>
> Each blue line shows results for a different model seeds. We now clarify this in the caption.
>
> > Where is the red line that is supposed to denote the ground truth?
>
> The red line is only visible in Fig 2e because that is the only plot where the ground truth has non-zero values. In Fig 2b, d, GT is not visible because the plot shows error relative to the GT, which is zero for the GT. We have now clarified this in the caption to avoid confusion.
>
> > Isn’t the error is larger in Dil-ResNet in fig 2b?
>
> It depends. During the time period covered by training, the Dil-ResNet’s Energy Field RMSE error is mostly lower than Athena++32 (Fig 2a, and interval [0-1] in Fig2b), even though Dil-ResNet’s also runs at the same 32^3 resolution. Just after the time period covered by training (1<t<2), the Dil-ResNet error is comparable, and long after (t>3) the time period covered by training, the Dil-ResNet error is higher. Dil-ResNet Energy Field RMSE error is in general higher than the higher-resolution Athena++ 64. This is not unexpected, as Athena++ 64 runs at higher resolution.
>
> However, as Fig 2d shows, Dil-ResNet’s error in the energy spectrum is much lower throughout. Fig 2e shows this advantage is due to better high frequency accuracy. This is also evident in Fig 2f-i, where the Dil-ResNet has high-resolution structure which is more analogous to the Athena++128 simulation, while the Athena++32 is much blurrier. Dil-ResNet spectrum error is even lower than that of the higher-resolution Athena++ 64.
>
> > What’s the point of t=1000 for a simulation that is already imprecise at t=1? Is it t=1 or t=1000 in figs 2f-2i? There are similar issues in other figs, like figs 4 and 5. It’s hard to show significant advantages with these low-accuracy simulations.
>
> We are confused by this question, but think it is because of our own notational error in the compressive turbulence case, as we had a few instances where we accidentally indicated milli-time units, rather than regular time units. For example, t=1000 should have been t=1, and $\Delta t$ = 32, should have been $\Delta t$=32e-3. We have corrected this in the text by removing usage of millitime units. For most analyses, learned models are rolled out from t=0 to t=1 (training) or to t=4 (generalization) in steps of $\Delta t$ =32e-3. Please let us know if this does not address the question.
>
> > As I said before, it would be more convincing if the authors can demonstrate some standard examples, such as Taylor Vortex, Karman Vortex, Frog-leaping, etc.
>
> We do have KS Equation as a standard benchmark. Our Incomp-2D domain involves complex vortex dynamics not unlike those of the simpler Taylor, Karman, and leapfrogging vortices. The 3D domains are more complex turbulence regimes which are actively studied in physics. We applied the same model across all these regimes because our intent was to cover a wide range of turbulence settings, including vortex dynamics.
>
> > For some turbulent flows that are chaotic themselves, the solutions at the resolution of 32 are meaningless, and the results are even worse than some large-eddy simulations.
>
> We are not entirely sure what the reviewer means by this, so let us know if we did not interpret this correctly. Yes, we agree, and it’s clear that the Athena++ 32 simulations are highly smoothed and hardly turbulent. However, our Dil-ResNet at 32^3 shows similar characteristics  as Athena++ 128^3 simulation (as characterized by the energy spectrum and other distributional metrics). This is one of the most interesting things we believe our work is showing.

---

> > ### Comment · Reviewer_HsnX · 2021-11-29
> > **Reply**
> >
> > Thank you for your reply. I marked up my score and I hope you can continue to polish the paper before submitting the final version.

---

### Official Review · Reviewer_zinf · 2021-11-02

**Correctness:** 3
**Technical Novelty And Significance:** 3
**Empirical Novelty And Significance:** 3
**Recommendation:** 6
**Confidence:** 3

**Main Review:**

This paper uses the existing modeling architecture on predicting turbulent flows. This paper shows several advantages of the proposed model: Compared with the classical solvers, using the same small resolution, the proposed model can produce more accurate results. It can also produce as accurate (or more accurate) results when using larger timesteps. The smaller grid, larger timestep, and the ability to utilize GPUs make this model more efficient than the classical solver.
This paper also compares different network architectures, which shows the best result is from using Dilated ResNet.
There are also certain discussions of the results, including the experiments of tuning training noise and temporal downsampling to solve the unstable issues, using different timesteps, evaluating longer rollouts, using different initial conditions, and using different box sizes.

Other questions:
For the CNN padding, can it deal with the scenarios of non-periodic and non-fixed boundary conditions?
When /Delta t changed, does the model need to be re-trained?
Using classical solvers, as the resolution increases, the results would gradually converge to the ground-truth value. Does this property hold true for the proposed model?
If we want to get more accurate results (for example, as accurate as using 256^3 Athena++, can the proposed model still out-performs the classical solver?

**Summary Of The Paper:**

This paper aims to learn a simulator that predicts large-scale turbulent dynamics of a known system. As shown in the results, on coarse grids, the proposed method predicts more accurately than the classical solvers, especially on preserving the high-frequency information.

**Summary Of The Review:**

Although using some existing learning models, there are certain contributions of this paper in applying the ml models on predicting the turbulent flows, which shows some advantages over the classical solvers.

===============
Updates: I thank the authors for the response. The response addresses my concern about the contributions of the paper. I would agree that  the empirical work of this paper is impressive. I decide to change my score towards acceptance.

---

> ### Author Response · Authors · 2021-11-19
> **Response to Reviewer zinf**
>
> Thank you for the thoughtful review. We address your questions below.
>
> > This paper uses the existing modeling architecture on predicting turbulent flows
>
> We just want to point out that while Dil-ResNet architecture is relatively simple and consists of previously published components (residual connections, dilated convolutions), the entire model (training noise, temporal coarsening, the way we combine residual connections, dilated convolutions in a U-shaped pattern) is novel, particularly for simulating turbulence. And despite the fact that Dil-ResNet is simpler and more generally applicable than the baselines, it quantitatively outperforms them.
>
>
> > For the CNN padding, can it deal with the scenarios of non-periodic and non-fixed boundary conditions?
>
> Yes. It should be able to deal with other boundary types (e.g. any combination of fixed (Dirichlet), fixed derivatives (Neumann), open boundaries (e.g. outflow)). It would just be necessary to (1) add an additional feature for each type of boundary, so the model can learn the dynamics of the boundary type, (2) impose loss on the boundary pixels to the extent that the values are not fixed by the boundary (e.g. outflow), and (3) enforce any boundary values/derivatives at each step of the trajectory (if applicable).
>
> > When /Delta t changed, does the model need to be re-trained?
>
> Yes. For example, in Figure 3 new models were trained for each $\Delta t$. Of course, for a model trained on some $\Delta t$, integer multiples of $\Delta t$ could be produced by applying the simulator iteratively. We have clarified this in the text.
>
> > Using classical solvers, as the resolution increases, the results would gradually converge to the ground-truth value. Does this property hold true for the proposed model? If we want to get more accurate results (for example, as accurate as using 256^3 Athena++, can the proposed model still out-performs the classical solver?
>
> It depends: Is the question (a) whether increasing the resolution of our model, which was trained at 32^3 to predict 128^3, would converge to the 256^3 solution? Or (b) whether if we trained a model at some resolution R^3 to predict 256^3, would it still be better at predicting 256^3 than a comparable classical solver run at R^3 (as our 32 → 128 results show)?
>
> For (a), no, we would not expect our models to generalize to resolutions it hadn’t been trained on because the neural networks within the architecture are being trained to process the structure at the scale experienced during training.
>
> For (b), yes, we don’t believe there’s anything privileged about the 128^3 ground truth, and we’d expect training on higher resolution targets should show similar advantages to our approach. Note, 128^3 is already quite high resolution, and computationally expensive to run.
>
> Regardless, this is probably outside the scope of our paper: the point is to approximate high resolution simulations with faster, lower resolution trained models, rather than to increase the resolution.
>
> > Although using some existing learning models, there are certain contributions of this paper in applying the ml models on predicting the turbulent flows, which shows some advantages over the classical solvers.
>
> Beyond the key question of the advantages of learned models over classical solvers, we want to emphasize the novelty of the approach. Our Dilated ResNet architecture is distinct from previous models applied to turbulence. As described above, it shared some building blocks (e.g. convolutional layers) with other work, but includes different elements like dilations and residual skip connections. The resulting model is simple and domain-general, and it still performs better than more complex and turbulence-specific architectures. We also tested these models in a wider range of settings and resolution manipulations than did previous work, and recommended training noise and temporal downsampling that improves performance across all architectures.
>
> We thank the reviewer for their time and consideration, and hope that our response has addressed their comments and made a good case for recommending acceptance.

---

### Official Review · Reviewer_1vW7 · 2021-11-02

**Correctness:** 3
**Technical Novelty And Significance:** 3
**Empirical Novelty And Significance:** 3
**Recommendation:** 8
**Confidence:** 3

**Main Review:**

I would like to applaud the authors' effort to perform extensive experiments on different systems with different models. I think the experimental results well support a general claim that a learned simulator is indeed promising for some turbulence phenomena.

That being said, I feel that the main claim of the paper is somewhat vague. Is it on the goodness of the proposed architecture, called dilated ResNets? If it is, then I would like to know more about the difference from the baseline methods. Since such a comparison result is shown only in a small subsection, as the authors also acknowledged ("... although most learned models were able to provide a good qualitative match ..."), I could hardly understand the overly notable superiority of the dilated ResNets. Figures B.3 and B.4 also provide some comparison, but to me, they were not really clear, either. Maybe it would be helpful if the authors can add some more explanations about the quantitative and qualitative differences between the architectures if any. If the authors think that the inter-architecture difference is not necessarily significant, then the main claim of the paper should be emphasized in a different way.

A small question on the training set. In the last two datasets, are the training sets generated with Athena++ 128 (and then downscaled)? If so, maybe it should be stated more clearly in the main text (sorry if there's already such a statement).

It would be kind if the semantics of "predicted Energy Field and the Log Energy Spectrum" are presented somewhere in the main text. I think many machine learners are not familiar with such notions.

On page 8, "We found that training with prevents this in some cases: ..." --> with what? Maybe a word is dropped.

**Summary Of The Paper:**

The authors evaluate some different architectures of neural nets for learning turbulence. They showcase the results for four different systems and in terms of several metrics. The main conclusion is that a learned simulator can indeed be good even for turbulence and possibly improved with additional mechanisms such as additional noise and constraints.

**Summary Of The Review:**

The work seems to be valuable in the sense that it conducts extensive experiments on simulator learning for turbulence. Meanwhile, the main claim of the paper is somewhat vague because the difference between the proposed architecture and the baseline architectures is not explained in full detail.

---

> ### Author Response · Authors · 2021-11-19
> **Response to Reviewer 1vW7**
>
> Thank you for the thoughtful review. We address your questions below.
>
> > I would like to applaud the authors' effort to perform extensive experiments on different systems with different models. I think the experimental results well support a general claim that a learned simulator is indeed promising for some turbulence phenomena. That being said, I feel that the main claim of the paper is somewhat vague. Is it on the goodness of the proposed architecture, called dilated ResNets? If it is, then I would like to know more about the difference from the baseline methods. Since such a comparison result is shown only in a small subsection, as the authors also acknowledged ("... although most learned models were able to provide a good qualitative match ..."), I could hardly understand the overly notable superiority of the dilated ResNets… Maybe it would be helpful if the authors can add some more explanations about the quantitative and qualitative differences between the architectures if any. If the authors think that the inter-architecture difference is not necessarily significant, then the main claim of the paper should be emphasized in a different way.
>
> Our main contribution is to show that learned simulators built from existing deep learning building blocks can model turbulence efficiently and accurately using coarsened grids. We evaluate performance across a variety of experimental domains and performance metrics. Regarding the proposed architecture, like many other models in the literature, Dilated ResNet is composed of existing deep learning building blocks (ResNets, dilations, U-shape). However, our model combines these in a way that is novel, is simpler than previous models, and achieves strongest quantitative results. Regardless, we view this work as not so much about introducing brand new approaches, as much as assessing the field, providing clear answers on resolution/accuracy tradeoffs, and offering prescriptions about models and training procedures to scientists and engineers.
>
> We’ll add more explicit clarification about qualitative and quantitative differences as suggested.
>
> > In the last two datasets, are the training sets generated with Athena++ 128 (and then downscaled)? If so, maybe it should be stated more clearly in the main Section  (sorry if there's already such a statement).
>
> Yes, this is correct. We have added “The training data is generated at high resolution and downsampled to coarsened resolutions. Spatial and temporal downsampling factors and additional details about the environments are shown in the Appendix.” to the Section 4.
>
> > It would be kind if the semantics of "predicted Energy Field and the Log Energy Spectrum" are presented somewhere in the main text.
>
> Thanks for the suggestion – added to Results Section:
>
> “We chose the energy field $E=\frac{1}{2}\rho v^2+\frac{3}{2}P$ as the quantitative metric of 3D turbulence for the main text, as it summarizes performance on all state variables.”
>
> “Log Energy Spectrum is computed by (1) taking the 3-D Fourier transform, (2) computing the amplitude of each component, (3) taking the histogram of the 3-D Fourier transform over frequency magnitude ($\sqrt{k_x^2+k_y^2+k_z^2}$) to get a 1-D spectrum, and (4) then taking the log.”
>
> > On page 8, "We found that training with prevents this in some cases: ..." --> with what? Maybe a word is dropped.
>
> Thank you, fixed (word was “noise”).

---

> > ### Comment · Reviewer_1vW7 · 2021-11-23
> > **Good empirical work**
> >
> > Thank you for the authors' rebuttal. The added perspectives and information will be highly useful to improve the paper. I marked up my score. I think this is a good collection of empirical results for turbulence modeling.
> >
> > Meanwhile, I would suggest weakening a bit the claim of the current paper about the superiority of dilated resnet. For example, the authors say in the introduction:
> >
> > > We implement this as well as a set of learned models from the literature that have been proposed for modeling turbulence, and show that Dilated ResNet outperforms the other, often more complicated models.
> >
> > By such statement, a reader would expect the superiority would be clearly shown, but I do not think the results are not so strong in this regard as mentioned in the initial review. Well, **maybe** the dilated resnet is not bad and even a bit better than others. Having said that, it is not overly apparent in the empirical results. I would say the authors can mention **possible** superiority of dilated resnet. Claiming superiority as if it was clear would be confusing.

---

### Decision · Program_Chairs · 2022-01-20

**Decision:**

Accept (Poster)

**Comment:**

The paper compared different architectures of deep neural nets for learning full 3D turbulence simulations.  On coarse grids, the proposed method predicts more accurately than the classical solvers, especially on preserving the high-frequency information.  The reviews think the paper is clearly written with strong experiments. Pls include the suggested references in the final version.